# Beyond Verifiable Rewards: Scaling Reinforcement Learning in Language Models to Unverifiable Data

**Yunhao Tang**

**Sid Wang**
Meta Platforms, Inc.

**Lovish Madaan**
Meta Platforms, Inc.

**Rémi Munos**
Meta Platforms, Inc.

## Abstract

We propose to scale RL to unverifiable data with a novel algorithm JEPO (Jensen's Evidence lower bound for Policy Optimization). While most prior effort on scaling RL for LLMs focuses on verifiable data where ground truth answers are typically short-form and can be matched easily, we investigate the case where such assumptions are less valid (e.g., when answers are long-form such as mathematical proofs). To scale RL training to unverifiable data with contemporary training constraints, we propose JEPO. JEPO applies Jensen's evidence lower bound, a pragmatic simplification of the evidence lower bound which views chain-of-thought as a latent variable in the generative process. We show that on verifiable datasets (math), JEPO is as effective as RL with verifiable reward; on semi-verifiable and unverifiable datasets (numina and numina-proof), JEPO improves on soft-match based evaluations compared to RL with verifiable reward which can only leverage a subset of the data source as well as test set likelihood evaluations.

## 1 Introduction

Reinforcement learning from verifiable reward (RLVR) has proved effective at endowing language models with capabilities beyond canonical pre-training and supervised fine-tuning [1, 2, 3, 4, 5, 6]. At its core, reinforcement learning (RL) allows for the training of chain-of-thought at scale, which can elicit significant performance improvements especially for reasoning intensive tasks [7, 8]. In the case of mathematical reasoning, it encourages step-by-step solutions that lead up to a final answer [9, 10], whose correctness can be verified to produce a reward signal for RL training.

However, a main limitation of current RLVR is the data source: verifiable rewards are mostly derived from datasets where ground truth answers are short-form and can be checked in relatively easy ways [4, 5, 6]. For example, most answers to popular benchmarks MATH [11] and AIME [12] are integers and can be verified using simple string match. Yet, for many data sources, the ground truth answers are less verifiable or even unverifiable by current standards. For example, for long-form data with the answer being the whole proof, its inherent correctness is hard to assess without human raters [13].

The boundary between verifiable and unverfiable data, though often blurry in practice, can be made actionable: we define data as unverifiable, if its ground truth answer cannot be verified with a reasonably simple automatic procedure. It is of interest to scale RL to such data sources, for a few notable practical reasons: (1) some data have inherently long answers which cannot be cast into short-form answers in a straightforward way; (2) data sources with long-form answers exist in abundance, and it is sub-optimal not to leverage such data for training. In this work, we seek to tackle the problem of scaling RL to these unverifiable data sources.

We propose JEPO (short for Jensen's Evidence lower bound for Policy Optimization), a novel RL algorithm that can equally post-train models on verifiable or unverifiable data. The design of the algorithm is inspired by a latent variable view of chain-of-thought [14, 15]. Different from prior work, we make use of *Jensen's evidence lower bound*, a novel pragmatic simplification of the full evidence lower bound [16, 17], named after Jensen's inequality [18]. Optimizing a simplified objective

39th Conference on Neural Information Processing Systems (NeurIPS 2025).

forgoes the complication of training expensive auxiliary models, making JEPO more suitable for contemporary large-scale training [19, 20].

The final algorithm consists of an hybrid online RL and SFT-like loss. When viewed as an alternative to online RL, the lower bound method does not require any external verifiable reward, forgoing the need for the ground truth to be easily verifiable. In sum, our technical contributions are as follows:

**Algorithm**   Followed by a brief background on latent variable modeling, we derive the Jensen's evidence lower bound in Section 3, the stochastic optimization algorithm. In Section 4, we show how its multi-sample extension [21] tightens the theoretical bound and alludes to better performance in practice. While JEPO works similarly as regular online RL baselines, we detail a few important aspects of its implementations in Appendix B.2.

**Theoretical Connections**   We offer an intriguing graphical model view of a few important algorithms connecting probabilistic inference with chain-of-thought optimization. See details in Appendix A. We further discuss the practical trade-offs and connections between JEPO and online RL algorithms. Due to space limits, such results are presented in Appendix D.

**Empirical Results**   Finally in Section 6, Section 7, and Section 8, we show that for verifiable data, JEPO is competitive compared to online RL with verifiable rewards. For semi-verifiable and unverifiable data, JEPO has performance advantage over online RL or SFT baselines. As a by-product, we also showcase the utility of generating chain-of-thought for long-form proofs, an observation that is interesting in its own right.

## 2   Reinforcement learning for language models

A language model can be understood as a policy $\pi_\theta$ in the context of reinforcement learning. Given a prompt $x$, the policy generates a response $y$, which then gets assessed by a human user. Usually, the objective is to optimize $\pi_\theta$ such that certain reward function $r(x, y)$ that captures human preference is maximized [22, 23]. Formally, consider the maximization problem

$$\max_\theta \mathbb{E}_{y \sim \pi_\theta(\cdot|x)} \left[ r(x, y) \right] - \beta \mathbb{KL} \left( \pi_\theta(\cdot|x), \pi_{\text{ref}}(\cdot|x) \right) \tag{1}$$

with a KL regularization that encourages $\pi_\theta$ to stay close to the reference policy. The reward $r(x, y)$ captures the human preference of response $y$ in response to prompt $x$ and can take various forms: for example, it can be extracted from human annotations [22, 24, 23], generated atuomatic feedback such as code execution [25]. We focus on a specialized setting where the reward is derived from access to certain *ground truth* of the problem.

### 2.1   RL from ground truth feedback

We focus on applications where the prompt $x$ typically specifies a question and there is an example of an desirable ground truth $a^*$. Such a formulation is applicable to mathematical reasoning [11, 26, 10] where $x$ is a question and $a^*$ is the ground truth answer. When the correctness of model generated answer $a$ can be easily verified against the ground truth $a^*$, a verifiable reward $r$ is available by matching $a^*$ against the answer $a$. As another example, when $a^*$ is a long-form proof, such a reward is not immediately available and such cases are considered less verifiable. In broader context, RLVR also includes code applications where the reward is computed via unit tests [25, 27]. We do not consider such use cases in this work.

### 2.2   Chain-of-thought

For aforementioned applications where the model is required to reason about the question $x$ and generate an answer $a$, getting the model to generate chain-of-thoughts - a sequence of reasoning steps $c$ leading up to the final conclusion [7, 8]. Henceforth, we can decompose the generation $y = (c, a)$ into a chain-of-thought $c$ and an answer $a$. The generative process for the response $y \sim \pi_\theta(\cdot|x)$ is made more concrete as $c \sim \pi_\theta(\cdot|x), a \sim \pi_\theta(\cdot|x, c)$. Given a prompt $x$, the intuitive role of chain-of-thought is such that it makes the *marginal* likelihood of the ground truth answer $a^*$ higher. As such, we can interpret chain-of-thought as a latent variable and formulate the optimization of chain-of-thought as latent variable modeling [15, 14].

# 3 A Jensen's lower bound for chain-of-thought as latent variable modeling

We start with the initial motivation to increase the marginal likelihood of the ground truth answer $a^*$ (i.e., the evidence) given the generative process with chain-of-thought

$$\max_\theta \log \pi_\theta(a^*|x). \tag{2}$$

Directly optimizing the log likelihood is not tractable because its gradient cannot be estimated via samples in an unbiased way (see, e.g., discussion on this in the probabilistic inference literature [17]). As the main contribution of this work, we propose a tractable lower bound objective by directly applying the Jensen inequality to lower bound the log likelihood

$$\log \pi_\theta(a^*|x) = \log \mathbb{E}_{c\sim\pi_\theta(\cdot|x)}\left[\pi_\theta(a^*|x,c)\right] \geq \underbrace{\mathbb{E}_{c\sim\pi_\theta(\cdot|x)}\left[\log \pi_\theta(a^*|x,c)\right]}_{\mathcal{L}_\theta(x,a^*)}, \tag{3}$$

where we exchange the order of the concave $\log$ function and expectation $\mathbb{E}\left[\cdot\right]$. There are conditions under which the lower bound $\mathcal{L}_\theta(x,a^*)$ is tight. For example, if all chain of thoughts $c$ in the support of $\pi_\theta(\cdot|x)$ induce the same probability of predicting the ground truth answer $\pi_\theta(a^*|x,c)$, i.e., $\pi_\theta(a^*|x,c) = \pi_\theta(a^*|x,c'), \forall c, c' \in \text{supp}\left(\pi_\theta(\cdot|x)\right)$. In practice when the optimization is approximate, such conditions are not likely to hold. As a result, there might be a gap between the lower bound and $\log \pi_\theta(a^*|x)$ and we will examine its empirical impact in practice.

The gap between the marginal log likelihood and the lower bound can be expressed as the KL divergence between $\pi_\theta$ and the posterior distribution [17]

$$\log \pi_\theta(a^*|x) - \mathcal{L}_\theta(x,a^*) = \mathbb{KL}\left(\pi_\theta(\cdot|x), p^{\pi_\theta}(\cdot|x,a^*)\right),$$

where $p^{\pi_\theta}(c|x,a^*) := \frac{\pi_\theta(a^*|x,c)\pi_\theta(c|x)}{\sum_{c'}\pi_\theta(a^*|x,c')\pi_\theta(c'|x)}$. The posterior defines a distribution over chain-of-thought, and effectively denotes how likely is the chain-of-thought $c$ given that the ground truth answer is $a = a^*$ and the prompt is $x$. For experienced readers, this lower bound is closely related to the evidence lower bound [28, 17], which we will elaborate more below.

## 3.1 Stochastic gradient estimate

The lower bound permits stochastic gradient estimates. Concretely, given samples from the current policy $c \sim \pi_\theta(\cdot|x)$, we can construct an estimate of $\nabla_\theta \mathcal{L}_\theta(x,a^*)$ as

$$\underbrace{\log \pi_\theta(a^*|x,c)\nabla_\theta \log \pi_\theta(c|x)}_{g_1} + \underbrace{\nabla_\theta \log \pi_\theta(a^*|x,c)}_{g_2}. \tag{4}$$

The gradient has two terms: $g_1$ is a REINFORCE gradient estimate with $\log \pi_\theta(a^*|x,c)$ as the reward function for sampled chain-of-thought $c$ [29]. The second gradient $g_2$ is reminiscent of a supervised learning loss that encourages the model to predict ground truth answer $a^*$ given sampled chain-of-thought $c$.

In practice, we can add a control variate to the REINFORCE gradient estimate to reduce variance. One option is to learn a prompt-answer dependent function [30]; another sample-based alternative is to generate $n$ i.i.d. chain-of-thoughts in parallel $c_i \sim \pi_\theta(\cdot|x)$, and construct leave-one-out control variates $v_i = \frac{1}{n-1}\sum_{j\neq i}\log \pi_\theta(a^*|x,c_j)$ [31, 32]. The overall gradient estimate is the average over $n$ samples:

$$\frac{1}{n}\sum_{i=1}^{n}\left[\left(\log \pi_\theta(a^*|x,c_i) - v_i\right)\nabla_\theta \log \pi_\theta(c_i|x)\right] + \frac{1}{n}\sum_{i=1}^{n}\left[\nabla_\theta \log \pi_\theta(a^*|x,c_i)\right]. \tag{5}$$

Note the control variates $v_i$s do not introduce any bias to the gradient estimate since they are statistically independent from $\nabla_\theta \log \pi_\theta(c_i|x)$.

**Connections to supervised fine-tuning** In the very special case where there is no chain-of-thought, the gradient estimate reduces to just the SFT part $\nabla_\theta \log \pi_\theta(a^*|x)$ which is effectively the supervised fine-tuning loss from prompt $x$ to answer $a^*$. Here, the key difference is that the loss $\pi_\theta(a^*|x,c_i)$ further conditions on the chain-of-thoughts $c_i$s whose distribution changes over time and introduce more non-stationarity to the optimization process.

# 4 Tightening the objective via multi-sample Jensen's lower bound

If the Jensen's lower bound is loose, it will induce a sizable discrepancy from the true objective of interest. We need strategies to tighten the lower bound for policy optimization.

A similarly simple yet a tighter lower bound alternative, is an extension to the multi-sample case [21]. Indeed, consider the $n$-sample Jensen's lower bound

$$\mathcal{L}_\theta^{(n)}(x, a^*) \coloneqq \mathbb{E}_{(c_i)_{i=1}^n \sim \pi_\theta(\cdot|x)} \left[ \log\left( \frac{1}{n} \sum_{i=1}^n \pi_\theta(a^*|x, c_i) \right) \right]. \tag{6}$$

Importantly, the $\log$ function is outside of the $n$-sample average to tighten the bound. It is straightforward to verify that $\mathcal{L}_\theta^{(1)}(x, a^*)$ recovers the Jensen's lower bound as defined before in Eqn (3). As shown in [21], the lower bound becomes tighter as $n$ increases $\mathcal{L}_\theta^{(n)}(x, a^*) \le \mathcal{L}_\theta^{(n+1)}(x, a^*)$ for any $n \ge 0$. As $n \to \infty$, the bound approaches the marginal likelihood $\mathcal{L}_\theta^{(n)}(x, a^*) \to \log \pi_\theta(a^*|x)$, the ultimate objective of interest, under certain regularity conditions on $\pi_\theta$.

To maximize the multi-sample lower bound $\mathcal{L}_\theta^{(n)}(x, a^*)$ with gradient ascent, we can construct the REINFORCE stochastic gradient estimate as follows,

$$\underbrace{\sum_{i=1}^n \log\left( \frac{1}{n} \sum_{j=1}^n \pi_\theta(a^*|x, c_j) \right) \cdot \nabla_\theta \log \pi_\theta(c_i|x)}_{g_1^{(n)}} + \underbrace{\nabla_\theta \log \frac{1}{n} \sum_{i=1}^n \pi_\theta(a^*|x, c_i)}_{g_2^{(n)}}. \tag{7}$$

Empirically, the first term $g_1^{(n)}$ tends to have high variance as $n$ increases [33], since the objective $\log \frac{1}{n} \sum_{j=1}^n \pi_\theta(a^*|x, c_j)$ correlates updates to all $n$ samples. Akin to before, we can introduce the leave-one-out control variate without incurring any bias for variance reduction [31, 32]

$$\sum_{i=1}^n \left( \log\left( \frac{1}{n} \sum_{j=1}^n \pi_\theta(a^*|x, c_j) \right) - \widetilde{v}_i \right) \cdot \nabla_\theta \log \pi_\theta(c_i|x)$$

where $\widetilde{v}_i = \log \frac{1}{n-1} \sum_{j \ne i} \pi_\theta(a^*|x, c_j)$. Note that the second term $g_2^{(n)}$, though can be estimated via random samples, is unlike a regular SFT loss since it is the log average of multiple probabilities, instead of the average of log probabilities. As $n \to \infty$, since $\log \frac{1}{n} \sum_{i=1}^n \pi_\theta(a^*|x, c_i) \to \log \pi_\theta(a^*|x)$, we see that at least conceptually $g_2^{(n)}$ can be understood as directly maximizing the marginal likelihood, where the average over probabilities effectively marginalize the chain-of-thought conditional distribution. As we will show in Section 6, multi-sample lower bound generally improves the single sample Jensen's lower bound. This means that tightened lower bound improve training objectives both in theory and in practice . Due to space limit, we detail important technical details of the practical implementation in Appendix B.

# 5 Connections to related algorithms and prior work

The JEPO algorithms bear close connections to a number of algorithmic alternatives, which we discuss in Appendix D due to space limit. See Algorithm 1 for the pseudocode of the full algorithm.

**Training with unverifiable data** A natural way to generalize RL training to unverifiable data is to make use of LLM feedback, e.g., *LLM-as-judge* uses LLM to assess the quality of the generated response [34, 35, 36]. However, despite its conceptual simplicity, LLM-as-judge might not produce reliable assessment for domain-specific or long-form data [10, 13]. When optimizing against judge scores, it is also more likely to over-optimize [37]. As a result, in this work we apply LLM-as-judge only for short-form evaluations and not for training.

Closely related to our work is the concurrent VR-CLI (verifiable reward with completion likelihood improvement) [38] where they apply log probs of golden generations as reward. Using our terminology, their approach resembles the first part of the gradient in Eqn (5) of the Jensen's evidence lower bound. Without a SFT-like component, their update does not optimize for the marginal likelihood only partially. JEPO also applies the multi-sample technique to tighten the lower bound, achieving better empirical performance, which we will demonstrate in Section 6.

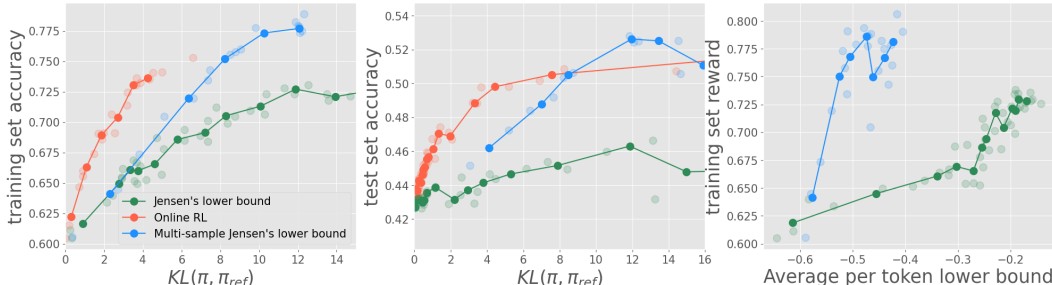

Figure 1: Short-form answer experiments with MATH. We compare three baselines: online RL with access to the oracle Sympy-based reward and JEPO. In the left plot, we monitor the reward on the training dataset. Online RL obtains the best training time trade-off, followed by multi-sample lower bound and the single-sample lower bound; In the middle plot, we monitor the evaluation on a test set during training. Multi-sample lower bound and online RL obtains similar performance; In the right plot, we graph training reward against the lower bound objectives, averaged over training tokens. The two signals bear positive correlations overall and multi-sample lower bound yields better correlations.

**Likelihood-based scoring** Prior work showcased the utility of Likelihood-based scoring in filtering of chain-of-thought [39, 40]. The algorithms mostly proceed in an iterative fashion akin to expectation-maximization [41], which in theory can also maximize the evidence of the desirable final answers. Complementary to such work, since we extend the training process to fully online RL settings, we forgo the need of variational posteriors which allows for training on unverifiable data at scale. To understand the limitations, we compare *reward-free* JEPO and RL with verifiable reward in Appendix D.4 where we highlight a genuine trade-off.

**Chain-of-thought as latent variable modeling** The idea of casting optimizing chain-of-thought as latent variable modeling is not new. Previously, [14] proposed an algorithm motivated by maximizing ELBO to tackle reasoning problems. Such an algorithm also draws close connections to prior work [42, 43, 44, 45] all of which resemble a hybrid offline-online RL training loop, where they alternate between sampling and filtering via a reward. They also have an interpretation as EM algorithmic variants [41].

Despite the appeal of a full ELBO formulation, it is rarely implemented in practice due to the requirement of learning the posterior distribution. Indeed, despite the formulation of [14] they ended up approximating the posterior with MCMC, which effectively made use of an explicit reward to filter samples. This also marks a key difference from our work - we do not apply any explicit reward scoring throughout our algorithmic design and practical implementation. In addition, [15] has proposed a more systemic hierarchical latent variable modeling view of chain-of-thought. Similar to our motive, [46] optimized an ELBO inspired objective for prompt selection, where they did not resort to an external reward. We discuss additional connections beten JEPO and full ELBO approach to chain-of-thought in Appendix D.1 and Appendix D.3.

**Evidence lower bound and RL** The connections between evidence lower bound and RL has been extensively studied in both the variational inference [47, 17] and RL community [48, 49]. In the RL literature, much of the variational inference view has been used to better interpret and improve existing algorithms with much focus on the goal-conditional problems, where a single reward is assigned at the end of a trajectory. Such a setting is quite akin to the RLHF case, where a sequence terminates with a single reward [50, 51, 52]. Our formulation also naturally incorporates the tighter multi-sample lower bound [21, 33] as special cases, which has seen little adoption in prior RL literature. In Appendix D.2, we discuss how JEPO relates to a variance reduced policy gradient algorithm akin to RLOO, defined as follows

$$\frac{1}{n}\sum_{i=1}^{n}\nabla_\theta \log \pi_\theta(c_i) \cdot (\pi_\theta(a^*|c_i) - \widetilde{w}_i) + \nabla_\theta \pi_\theta(a^*|c_i),$$

where $\widetilde{w}_i = \frac{1}{n-1}\sum_{j\neq i}\pi_\theta(a^*|c_j)$ is the leave-one-out baseline akin to similar constructs in JEPO.

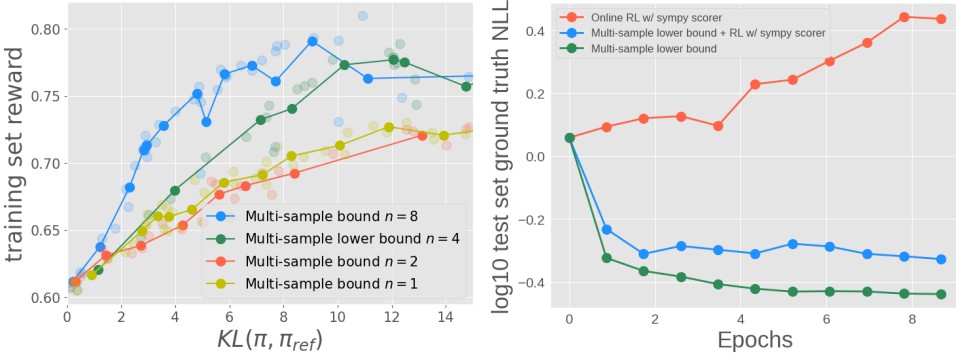

(a) Ablation with number of samples $n$ for JEPO    (b) Ablation with mumina test set NLL

Figure 2: Figure (a): Ablation of number of samples $n$ for multi-sample lower bounds. As we increase the number of samples, the multi-sample lower bound seems to further improve the training-time efficiency. This corroborates the theoretical insight that as $n$ increases, the multi-sample lower bound objectives become tighter. Figure (b): Test set proxy NLL evaluation for training on the numina dataset. We evaluate the proxy NLL of the trained models on the numina test set, approximated with $n = 4$ samples lower bound defined in Eqn (8). Both JEPO and the combined algorithm sees improvement in the NLL (lower the better), while online RL does not improve on test set NLL. This hints at different solutions found by the online RL and JEPO algorithms, despite similar improvement trend in the sampling based evaluations.

## 6 Experiments with verifiable data

We start by comparing JEPO against RL baselines on verifiable data. We focus on the mathematical reasoning dataset MATH [11] where the prompt $x$ asks the model a mathematical question with a short form answer $a^*$. We mainly focus on the two algorithmic variants proposed in this work: the JEPO defined through the gradient estimate in Eqn (4) as well as its multi-sample variant Eqn (7). As a strong baseline, we consider the online policy gradient RL algorithm which applies Sympy [53] to automatically match the answers. The RL algorithm applies leave-one-out for variance reduction, as is commonly practiced [54, 2]. Our main experiments are based on the 8B and 70B model from the Llama 3 model family [55]. All algorithmic variants apply identical hyper-parameters which we detail in Appendix B.

We highlight again that the RL baseline is at an advantage in this setting, since the reward is fairly *accurate* and is itself being used as evaluation signals too [56]. We do not compare with other baselines developed in prior work (e.g., [14]) as they can be interpreted as variants of online RL algorithms with certain low-level implementation differences.

### 6.1 Comparion on MATH

During RL training, we use a reward of $r = 1$ when there is an answer match and $r = 0$ otherwise. Note that JEPO does not require access to such a reward, but we monitor the reward scores during training. Figure 1 left plot shows the training performance of all baselines. For the x-axis, we use the KL divergence $\mathbb{KL}(\pi_\theta, \pi_{\text{ref}})$ calculated over the training set. Following the practice in [37], we adopt the KL divergence as a certain measure of the optimization budget that the algorithm has consumed. Note that here all experiments are run with the same regularization coefficient $\beta = 10^{-3}$ since it achieves a good trade-off for all algorithmic variants over all.

**Training performance**    Figure 1 left plot shows that online RL achieves a good KL-performance trade-off on the training set. This is probably not a big surprise since online RL optimizes for the very same objective that we monitor here. In the meantime, JEPO enjoys reasonable performance: as the policy deviates from the reference policy, the reward performance improves despite not explicitly training for it (in theory JEPO optimizes for a hard string match rather than Sympy match). (2) the multi-sample JEPO obtains noticeably better performance than the one-sample lower bound baseline, despite using the same $n = 4$ generations per update.

**Evaluation**    Figure 1 middle plot shows the evaluation performance on an held-out test set. We note that the reward on the training set is higher than the test set, because the model has been SFT'ed on on the training prompts. For evaluation, observe that the multi-sample lower bound method obtains

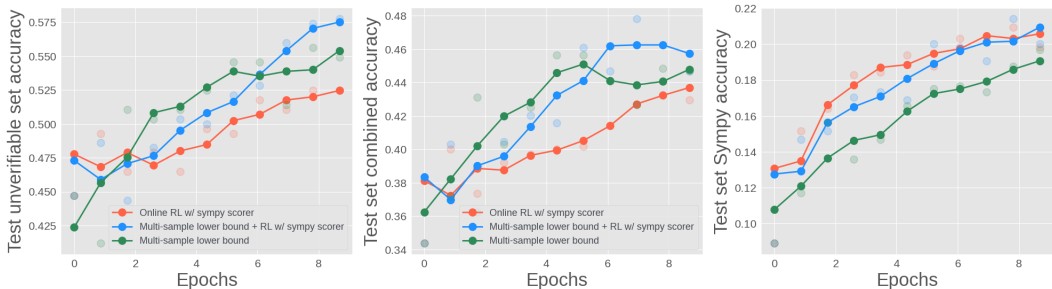

Figure 3: Evaluation comparison of training 70B models on semi-verifiable Numina dataset. We show evaluation results during the course of training. Left plot shows the combined accuracy on the unverifiable subset (about 40%) of the test set; middle plot shows the combined accuracy on the full test set; right plot shows the Sympy score on the full test set. While JEPO progresses more slowly on the Sympy scores compared to online RL, it gains on the combined accuracy; the combined algorithm seems to achieve the best of both worlds.

similar performance as online RL, despite being outperformed during training. We conjecture that this is because online RL tends to overfit the training prompts more significantly, producing a high training reward that does not transfer as well to the evaluation time. This shows that even without training on the reward signal explicitly, JEPO can obtain a similar evaluation performance as RL.

**Statistical correlation between objectives** Figure 1 right plot graphs the training time reward against the lower bound objectives. If we consider the training reward as a ground truth objective to optimize for, we see that the multi-sample lower bound displays a stronger correlations between the surrogate objective and the ground truth. Due to space limit, we show a comprehensive set of ablations in Appendix C.1.

## 7 Experiments with semi-verifiable data

We now consider semi-verifiable data where a good proportion of the dataset contains answers which are not easily verifiable. We focus on a post-processed Numina dataset [57] where prompts are mathematical questions and ground truth answers are partly verifiable. For instance, one example of the ground truth answer is the whole expression: $\forall x \in \mathbb{R}, x^2 + (a-1)x + 1 \geq 0$. Given a model generated answer, it is hard to verify whether it is equivalent to the above expression without case-specific parsing. See Appendix B for details on how we post-process the dataset.

**RL baseline and reward** For the RL baseline, we apply the Sympy reward as introduced in the previous section. Because the dataset contains answers which are hard to verify, the reward is effectively only applicable to a subset of the data. The default training set contains about 22k examples. We estimated at least 40% of such examples cannot be verified by the automatic scorer. We consider online RL with such reward as a baseline, as it has access to a highly specialized verifiable reward but only applicable to a subset of the data.

**Combining JEPO and RL baseline** We also compare with an algorithm that combines the loss function of JEPO and RL baseline with the Sympy reward. When we sample $n$ generations from a single prompt, and if none of the generation obtains a positive score (note this does not mean that the example is necessarily unverifiable), we apply the JEPO loss; otherwise, we apply the baseline RL loss. This allows for a dynamic combination of two losses, and still leverages the whole dataset.

**Evaluation** We have an held-out Numina test set consisting of 640 examples. The test set contains both verifiable and unverifiable examples, which we evaluate in two ways: (1) Sympy reward $r_{\text{sympy}}(a, a^*) \in \{0, 1\}$, which generally underestimates the true accuracy when ground truth is semi-verifiable; (2) Sympy combined with LLM-as-judge $r_{\text{combined}}(a, a^*)$, which combines two sources of scores $r_{\text{combined}}(a, a^*) := r_{\text{sympy}}(a, a^*) + r_{\text{llm}}(a, a^*)\mathbb{1}_{\{r_{\text{sympy}}(a,a^*)=0\}}$. The LLM-as-judge score $r_{\text{llm}}(a, a^*)$ is also binary: it is based on a 5-time majority voted decision of a prompted 70B instruction-tuned model [55]. Though imperfect, we observe that LLM-as-judge reasonably mitigates some false negatives caused by rigid Sympy scoring. Importantly, we reiterate that we do not train on such combined scores - they are used for evaluations only.

## 7.1 Comparison on Numina

Unless otherwise stated, we will experiment with the multi-sample algorithm given its performance gains in Section 6. Below, Figure 3 shows the evaluation performance comparing the RL baseline, JEPO and their combined algorithm. Since the Numina dataset is more challenging, we experiment throughout with 70B models.

**Sympy scoring evaluation**  Figure 3 right plot shows the evaluation accuracy using the Sympy score. Overall, all algorithms make steady progress as the training progresses. However, since online RL baseline trains with the same reward signal, it achieves slight acceleration compared to JEPO. The combined algorithm achieves a similar rate of progress with the Sympy scores on the test set.

Due to the abundance of symbolic expressions as ground truth in the Numina dataset, here the Sympy reward is a much more specialized scoring method than e.g., string match compared to the MATH case. This partly explains why the online RL baseline is competitive, as it trains on the same signal.

**Combined scoring evaluation**  Figure 3 left plot and middle plot shows the combined accuracy which alleviates some false negatives due to the Sympy scoring. As seen from the overall metrics, the accuracy increases by about 25% compared to the Sympy scores. The left plot shows the performance on the unverifiable test subset (40% of the test set) while the middle plot shows the full set. We observe that both JEPO and the combined algorithm achieves faster rate of progress and asymptotes to slightly better performance than the online RL baseline with this combined metric, especially on the unverifiable subset. Interestingly, note that by training on verifiable rewards, online RL can also make progress on the unverifiable test set.

Though the Sympy scoring is quite specialized, it is only applicable to a subset of the full Numina training set. Meanwhile, JEPO can leverage the full dataset, despite with less specialized signals. The combined algorithm seems to achieve the best of both worlds.

## 7.2 Ablation study

We carry out additional ablations to better understand the performance difference.

**Assessment with set negative likelihood: lower the better**  We further evaluate the proxy negative log likelihood (NLL) that the trained model produces on test set, computed via the $n$-sample lower bound [21, 40]

$$\text{proxy-NLL}(\pi_\theta) = -\mathbb{E}_{(c_i)_{i=1}^n \sim \pi_\theta(\cdot|x),(x,a^*)\sim\mathcal{D}_{\text{test}}}\left[\log\left(\frac{1}{n}\sum_{i=1}^n \log\pi_\theta(a^*|x,c_i)\right)\right]$$

Figure 2(b) shows such proxy NLL during training, where we see a different pattern for the online RL baseline and JEPO. For JEPO, the proxy NLL decreases over time. We expect such a result because JEPO optimizes for the same objective on the training set, and before overfitting, we expect improvement on the test set. Meanwhile, maybe surprisingly, online RL does not make progress on the test set NLL. The combined algorithm is in between the two extremes.

There are maybe good reasons for online RL not to make progress on test set NLL. Particularly, for each ground truth answer in the dataset $a^*$, the Sympy scorer defines a sizable collection of correct answers $\mathcal{A} = \{a : r_{\text{sympy}}(a, a^*) = 1\}$ whose aggregate probability $\pi_\theta(\mathcal{A}|x)$ increases under online RL (evidenced by test set accuracy improvement in Figure 3 right plot). In other words, online RL might not improve the proxy NLL of a particular $a^*$ inside $\mathcal{A}$. The above observation means that the policy found by online RL and JEPO can produce different answers to the same question. This is related to the model calibration issue for RL post-training in general [20].

**Comparison with SFT baseline on golden chain-of-thought**  To assess another option to improve semi-verifiable performance, we carry out another comparison against a SFT baseline, which trains on the golden chain-of-thought dataset [57]. We observe performance improvements across evaluation metrics as well, though generally underperforming RL. See Appendix B for full results.

## 8   Experiments with unverifiable data

Finally, we experiment with unverifiable data, where the full dataset has long-form ground truth and cannot be easily verified with hard-coded programs. We consider a post-processed Numina-proof, extracted from the original Numina dataset where all ground truth answers are full proof. The proof

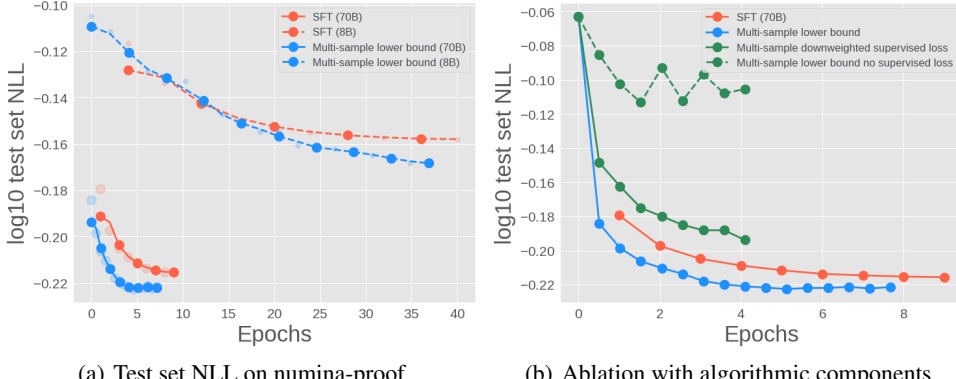

| (a) Test set NLL on numina-proof | (b) Ablation with algorithmic components |

Figure 4: Figure (a): Test set proxy NLL evaluation for training on the unverifiable Numina-proof dataset. For JEPO outperforms the SFT baseline at the same data budget (measured in epochs) and achieves asymptotically better test performance. Figure (b): Comparison of different baselines on numina-proof test set NLL, across various algorithmic variants, with the 70B model. We observe that the supervised component of the JEPO loss plays a key role at learning efficiency and achieving good asymptotic performance.

often contains multiple sentences of paragraphs, without a final short-form answer as in MATH or the verifiable subset of Numina.

**Baselines and evaluation**  Since the ground truth is long-form and cannot be verified easily, we do not have a RL baseline with verifiable reward. Instead, SFT on the raw dataset $(x, a^*)$ is a reasonable baseline. Through a few ablations, we also compare with methods akin to VR-CLI [38], which corresponds to the REINFORCE part of the single-sample lower bound gradient in Eqn (5). We evaluate NLL on the test set, akin to the ablations in Section 7. We do not carry out sampling based evaluations as long-form answers are hard to assess even for frontier models [13].

## 8.1 Comparison on Numina-proof

As main experiments, we compare JEPO with SFT. Note that we always started with instruction-tuned models [55] and the SFT baseline can be understood as a continued SFT. We show the curve after an initial transient phase where the test set NLL drops significantly for all runs, which can be attributed to that the modes learn to format answer correctly. Figure 4(a) shows the test set NLL comparison between SFT and JEPO, with both 8B and 70B models. At both scales, JEPO outperforms SFT with test set NLL at the same training data epoch. Also, JEPO is asymptotically better than SFT.

## 8.2 Ablation study on the importance of JEPO supervised loss

To better understand the role of different algorithmic components, we compare with additioanl baselines: recall that JEPO update contains two parts: a REINFORCE component, whose single-sample variant is akin to VR-CLI [38]; and a supervised loss component. We compare with a variant where the supervised loss is down-weighted ($\beta_{\text{sup}} = 0.01$) and another where it is removed ($\beta_{\text{sup}} = 0$).

Figure 4(b) shows the comparison on the test set NLL. We see that by downweighting the supervised loss, JEPO makes much less progress on the test NLL given the same training epochs. Specifically, when the supervised loss is removed ($\beta_{\text{sup}} = 0$), test NLL also seems to plateau at a worse level. Interestingly, this contrasts the observation in MATH experiments (Section 6) where small values of $\beta_{\text{sup}}$ works better. We speculate that the key difference is that nature of the chain-of-thoughts differs: for MATH, the chain-of-thought details solution steps and a final answer can be readily inferred. For long-form data, the chain-of-thought is a high-level outline, and it still takes extra effort to produce the full answer (e.g., proof), hence the importance of the supervised loss.

# 9 Limitations and future work

Possible directions for future research include studying the impact that various loss components (e.g., the REINFORCE and the supervised loss) have on overfitting; more organic ways to combine verifiable rewards and JEPO losses; ways to scale such methods, to more general purpose data (e.g., in the form of meta-thought [1, 58] or to pre-training [40].

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

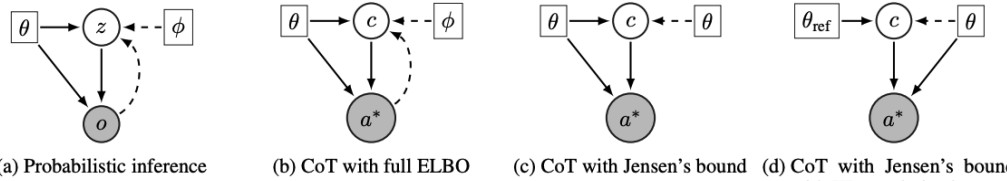

(a) Probabilistic inference     (b) CoT with full ELBO     (c) CoT with Jensen's bound    (d) CoT with Jensen's bound with KL regularization

Figure 5: Graphical models for various algorithmic formulations discussed in this work. Solid lines represent generative models and dashed lines represent inference models. Circles represent random variables and squares represent parameters. Shading indicates that the random variable is observed, and is used for providing feedback for the learning process. For CoT optimization, $a^*$ is a simplified notation for the binary optimality variable $\mathbb{1}_{\{a=a^*\}}$ from the random variable $a$.

## A    Review of the graphical model perspective

We make a more extended discussion about the graphical model shown in Figure 5.

**Probabilistic inference with a learnable prior**    Figure 5(a) shows the generic structure for probabilistic inference with a learnable prior, with latent variable $z$ and observable $o$. Here, $\theta$ controls both the prior and observation generation:

$$z \sim p_\theta(\cdot), o \sim p_\theta(\cdot|c).$$

The inference parameter $\phi$ denotes a the posterior inference distribution $q_\phi(z|o)$ that seeks to approximate the true posterior $p_\theta(z|o) := \frac{p_\theta(c)p_\theta(o|c)}{\sum_{c'} p_\theta(c')p_\theta(o|c')}$. Together, they can form an ELBO that lower bounds the marginal log likelihood [17]

$$\log p_\theta(o) \geq \underbrace{\mathbb{E}_{z \sim q_\theta(\cdot|o)} \left[ \log p_\theta(z|o) + \log \frac{q_\phi(z|o)}{p_\theta(z)} \right]}_{\mathcal{L}_{\theta,\phi}(o)}.$$

The right hand side $\mathcal{L}_{\theta,\phi}(o)$ can be optimized via stochastic gradient descent on the joint variable $(\theta, \phi)$. The lower bound is tight when the inference distribution is exactly the posterior $q_\phi(z|o) = p_\theta(z|o)$. A learnable prior refers to the fact that the prior distribution over latent $p_\theta(z)$ depends on $\theta$ too, while in much of the prior literature is is kept constant [59, 17]. For the transition from generic probabilistic inference to our use case, a learnable prior is also fundamentally important.

**Chain-of-thought with full ELBO**    Figure 5(b) shows a direct mapping of the probabilistic inference structure to the case of optimizing chain-of-thought. Here, the chain-of-thought $c$ is the latent variable and the ground truth answer $a^*$ is the observable. A more precise mathematically definition would be to consider yet another binary optimality variable $O = \mathbb{1}_{\{a=a^*\}}$ that determines whether the random variable answer $a$ is optimal. Here, we directly replace it with $a^*$ for notational simplicity.

If we further introduce a general conditional dependency on the prompt $x$, we arrive at the lower bound defined in Eqn (3)

$$\log \pi_\theta(a^*|x) \geq \underbrace{\mathbb{E}_{c \sim q_\theta(\cdot|x,a^*)} \left[ \log \pi_\theta(a^*|x,c) - \log \frac{q_\phi(c|x,a^*)}{\pi_\theta(c|x)} \right]}_{\mathcal{L}_{\theta,\phi}(x,a^*)}.$$

**Chain-of-thought with Jensen's lower bound**    In Figure 5(c), we replace the variational posterior $q_\phi$ by the prior distribution itself $\pi_\theta$. As discussed in the main paper, this looses the lower bound but make the optimization objective much simpler. See detailed derivations in Section 3. We see there there appears to be a duplicated arrow that goes from $\theta$ to the latent variable $c$. We make such duplication to distinguish between the inference distribution (dashed arrow) and the generative distribution (solid arrow); in this particular case, we deliberately make the two distributions identical.

**Jensen's lower bound with regularization**    Finally, Figure 5(d) presents the graphical model for the case where a KL regularization is added to the Jensen's lower bound (see Lemma D.6 for formal statements). In this case, the generative prior distribution is computed from the reference policy $\pi_{ref}$ parameterized by $\theta_{ref}$ which is kept fixed during training, while the rest of the distributions are still parameterized by $\theta$.

**Algorithm 1** JEPO: chain-of-thought optimization with Jensen's lower bound (or its multi-sample extension)

---
1: **INPUT** policy $\pi_\theta$
2: **while** $t = 0, 1, 2...$ **do**
3:     (i) For each sampled prompt $x$, collect $n$ generations $(y_i)_{i=1}^n$ and extract their corresponding chain-of-thoughts $(c_i)_{i=1}^n \sim \pi_\theta(\cdot|x)$.
4:     (ii) Evaluate $\pi_\theta(a^*|x, c_i)$ with a forward pass; calculate gradients $\nabla_\theta \log \pi_\theta(c_i), \nabla_\theta \log \pi_\theta(a^*|x, c_i)$ with backprop.
5:     (iii) Update $\theta$ with $n$-sample average of gradient estimate Eqn (4) or its multi-sample variant Eqn (7).
6: **end while**

---

# B  Hyper-parameters and experimental settings

We experimented with the Llama 3 model of size 8B and 70B. All experiments are conducted with identical hyper-parameter settings: we always apply a batch size of $B = 64$ prompts per update, and sample $n = 4$ distinct generations per prompt. All training and evaluation sampling are conducted at a temperature of $\tau = 1$ and with top-p $= 1$.

We train on the MATH training set with 7500 examples and evaluate on the test set with 2500 examples. A supervised fine-tuning on the training set is conducted to warm up the RL training, hence the gap between training and test set accuracy.

For both training and evaluation, we provide system instructions that ask the model to generate a response with step-by-step solution, followed by a final conclusion phrased as *the final answer is* followed by the answer predicted by the model. This is consistent with the prompt structure discussed for Llama models [55, 56].

All experiments are conducted with an entropy regularization coefficient $\beta > 0$ which we have ablated in the main paper.

## B.1  Dataset post-processing

We use unverifiable proofs data from Numina 1.5 [60] for our experiments. We clean and filter the questions and their corresponding solutions using some simple regex heuristics. For example, we replace leading blanks, markdown headings like ##, prefixes like "Problem:" and "Solution", letter-digit combinations like "A1" / "G5" / "ROU", and trailing dots and blanks. After cleaning, we have 58088 proofs from the Numina dataset.

## B.2  Important technical details of JEPO implementations

We detail the implementation details of the JELB-RL algorithm. We highlight a few key technical elements in the implementation, which we have found to be important in getting the best performance.

**Formatting penalty**   We find it useful to have an additional RL loss with the reward as $r_{\mathrm{reg}}(x, y) = -p$ if $y$ does not follow the formatting requirement (that the identifier phrase *the final answer is* is in $y$) and zero otherwise. We find that this generally helps stabilize the training process. This is especially useful for small models (8B) where under temperature sampling, it can often not follow instructions strictly. For large models (70B), we also found that its formatting might be inconsistent after multi-epoch training. We find a value of $p = 10$ suffices while smaller values tend to make the training less stable due to weaker penalties.

**Per-sequence log probabilities**   During the *log-ave-exp* operation that defines the lower bound in Eqn (3), it is important to apply the per-sequence log probs without average over the sequence length. Concretely, the bound is calculated as follows

$$\log \left( \frac{1}{n} \sum_{j=1}^n \sum_{t < |a^*|} \pi_\theta(a_t^*|x, c_j, a_{<t}^*) \right)$$

where $|a^*|$ denotes the sequence length of the ground truth $a^*$. It is important *not* to average the sequence level log probs $\sum_{t < |a^*|} \pi_\theta(a_t^*|x, c_j, a_{<t}^*)$ with a factor of $1/|a^*|$ as suggested in other contexts [61, 2], as it can modify the objective landscape significantly especially when $|a^*|$ is large.

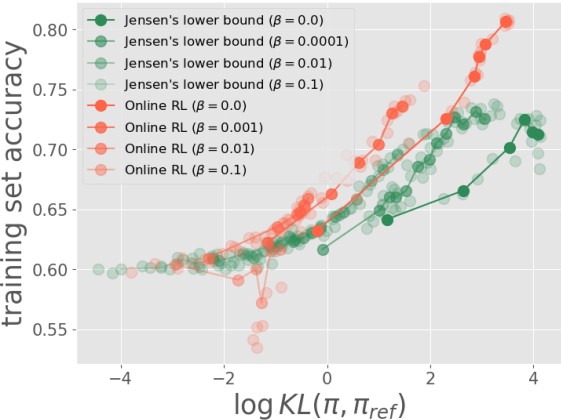

Figure 6: Ablation of regularization coefficient $\beta$. As $\beta$ increases, all algorithmic variants seem to obtain better efficiency in the training performance-KL divergence trade-off. However, strong regularization also prevents the policy from deviating much from the reference policy, preventing bigger training improvements.

**Advantage normalization** Both the baseline RL and the JEPO apply advantage post-processing, following common practice [30, 62]. For example, in the multi-sample JEPO, the advantage for the $i$-th generation is

$$A_i = \log\left(\frac{1}{n}\sum_{j=1}^{n}\pi_\theta(a^*|x,c_j)\right) - \widetilde{v}_i.$$

A further normalization is applied to the advantage $\widetilde{A}_i = \text{clip}(A_i/\text{std}\,(A)\,, -1, 1)$ such that the outcome $\widetilde{A}_i$ is applied in the actual update. Advantage normalization is especially important for JEPO because its raw advantage takes a wider range of numerical values.

**Weighted supervised learning loss** We also introduce a weighting coefficient for the supervised loss $\beta_{\text{sup}}$ useful for ablations. We find that small values $0 \sim 10^{-2}$ tends to work for short-answer applications (e.g., MATH) while a large value $1$ is important for semi long-form data (e.g., numina and numina-proof).

**KL-regularization** In our early investigation, we found it useful to have a KL regularization at a very small coefficient $\beta = 10^{-3}$. The regularization helps prevent formatting collapse, and also prevents the policy from drifting too much in case the updates are noisy [24, 23].

Put together, given $n$ samples, the JEPO update is

$$\frac{1}{n}\sum_{i=1}^{n}\left(\left(\widetilde{A}_i + \widetilde{A}_i^{(\text{ref})}\right)\nabla_\theta\log\pi_\theta(a_i|x,c_i)\right) + \beta_{\text{sup}}\nabla_\theta\log\left(\frac{1}{n}\sum_{i=1}^{n}\pi_\theta(a^*|x,c_i)\right) - \beta\nabla_\theta\mathbb{KL}\left(\pi_\theta(\cdot|x),\pi_{\text{ref}}(\cdot|x)\right),$$

where $\widetilde{A}_i^{\text{ref}}$ is the normalized advantage for the formatting penalty. The normalization makes it such that the ultimate update optimizes for a lower bound more akin to the weighted lower bound [63] though the underlying algorithmic motivations differ.

The lower bound loss is applied only to generations with correct format, otherwise, the loss is masked out. Also, we find that the sequence level normalization with a factor of $1/\left(|c_i| + |a^*|\right)$ or $1/|c_i|$ does not make a significant difference [2, 64].

## C  Additional ablations

We now provide ablation results on a few important dimensions of the algorithm.

### C.1  Ablations for verifiable data

We detail additional ablations for the verifiable data.

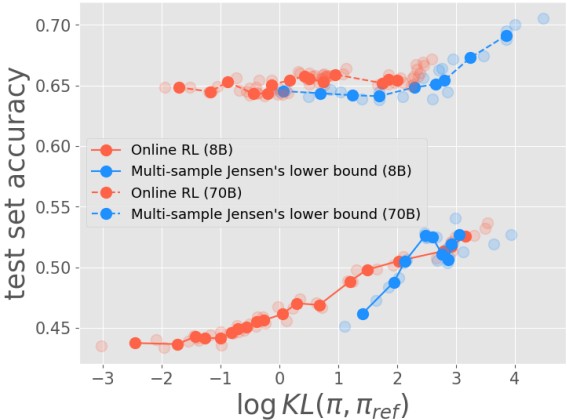

Figure 7: Ablation of model size (8B vs. 70B). We find that the multi-sample JEPO is fairly competitive against the online RL algorithm in the 70B scale. Both algorithm traces out a similar KL-performance trade-off, with multi-sample JEPO obtaining a slightly better performance given a similar compute budget as online RL.

**Multi-sample ablation on sample size** $n$ We ablate on the number of sample $n$ used for constructing per gradient update. In theory, as $n$ increases, the multi-sample lower bound becomes tighter and asymptotically approaches the marginal likelihood objective (which is equivalent to the RL objective). We vary the sample size $n \in \{1, 2, 4, 8\}$ and compare the performance. Figure 2(a) shows that as $n$ increases, the algorithm becomes more KL-efficient: with a fixed budget on KL, the model obtains better performance. Intriguingly, we also observe a training performance akin to reward over-optimization [37] - as the optimization progresses, the training reward drops slightly (for blue curve). We can interpret this as the result of the fact that JEPO does not optimize for the same indicator matching function as the reward we monitor.

**Regularization ablation** We investigate the impact of the regularization coefficient $\beta \in \{0, 10^{-3}, 10^{-2}, 10^{-1}\}$. Figure 6 shows the training performance of the single-sample lower bound vs. online RL. One observation is that as $\beta$ increases, the trade-off efficiency for both algorithms improves - however, in general the algorithm also makes less deviation from the reference policy, hence leading to less improvement for a fixed training steps.

**Scaling up model size** Since the multi-sample JEPO appears more competitive, we compare it against the online RL in the 70B case. Figure 7 shows that the JEPO obtains competitive performance against online RL in terms of the KL-performance trade-off. With roughly the same amount of compute budget, we find that the JEPO seems to drift further from the reference policy, hence extending the trade-off curve to a performance of 70% test set accuracy, which outperforms online RL modestly.

**Supervised loss** We find that a low value of $\beta_{\text{sup}}$ generally works better for the JEPO algorithms. The speculation is that when $\beta_{\text{sup}}$ is large, the supervised loss encourages the policy to place weights on the ground truth $a^*$ despite that the chain-of-thought $c$ has low quality. This leads to overfitting the training set, in a more severe way than online RL.

### C.2 Ablations on semi-verifiable data

**Comparison with SFT baseline on golden chain-of-thought** See ablation results in Figure 8. A few observations are in order: (1) SFT generally is not as good as the RL jobs, but it improves over time as we train more; (2) There is an initial drop in performance, which can be explained by the fact that the golden chain-of-thought does not conform to the familiar "step-by-step" that the starting model has been post-trained with [55]. Through SFT, the model needs to unlearn the step-by-step format and learns the more freeform hybrid format in the golden chain-of-thought data; (3) Asymptotically, SFT performs lower than RL runs.

## D JEPO's connections with alternative algorithmic variants

We discuss detailed connections between JEPO and alternative algorithmic formulations below.

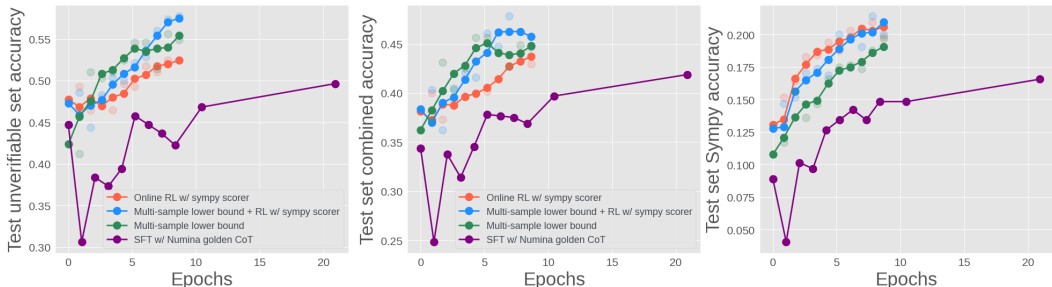

Figure 8: Additional comparison against a SFT baseline which trains on golden chain-of-thought data from the numina dataset. We show that the SFT baseline also improves upon various metrics, despite generally underperforming RL algorithms.

### D.1  Evidence lower bound

The evidence lower bounds (ELBO) [16, 28, 21] control for the tightness of the lower bound with an inference distribution $q_\phi(c|x, a^*)$ which defines a distribution over chain-of-thoughts. The lower bound takes the following form

$$\mathcal{L}_{\theta,\phi}(x, a^*) = \mathbb{E}_c \left[ \log \pi_\theta(a^*|x, c) - \log \frac{q_\phi(c|x, a^*)}{\pi_\theta(c|x)} \right], \tag{8}$$

where the expectation is under $c \sim q_\phi(\cdot|x, a^*)$. It lower bounds the marginal log likelihood $\mathcal{L}_{\theta,\phi}(x, a^*) \leq \log \pi_\theta(a^*|x)$ and it is tight if and only if the inference distribution equals the posterior distribution $q_\phi(c|x, a^*) = p^{\pi_\theta}(c|x, a^*)$. Since ELBO is a function of both policy parameter $\theta$ and inference distribution parameter $\phi$, we can optimize both with stochastic gradient estimates: given a chain-of-thought sample $c \sim q_\phi(\cdot|x, a^*)$,

$$g_\theta = \nabla_\theta \log \pi_\theta(a^*|x, c) + \nabla_\theta \log \pi_\theta(c|x),$$

$$g_\phi = \nabla_\phi \log q_\phi(c|x, a^*) \left( \log \pi_\theta(a^*|x, c) - \log \frac{q_\phi(c|x, a^*)}{\pi_\theta(c|x)} \right)$$
$$\quad - \nabla_\phi \log q_\phi(c|x, a^*).$$

Juxtaposing the form of the gradient here and the gradient to the Jensen's lower bound defined in Eqn (4), we observe that the inference distribution gradient $g_\phi$ bears resemblance to the REINFORCE gradient; while the policy distribution gradient $g_\theta$ bears resemblance to the SFT gradient. In fact, we can show that under the special parameterization $q_\phi(c|x, a^*) := \pi_\theta(c|x)$, the two gradients are exactly equivalent. Such an observation is stated formally below.

**Lemma D.1.** (**Jensen's lower bound as a special case of ELBO**) When $q_\phi(c|x, a^*) := \pi_\theta(c|x)$, ELBO is equivalent to the Jensen's lower bound $\mathcal{L}_{\theta,\phi}(x, a^*) = \mathcal{L}_\theta(x, a^*)$ gradient esimtates are equivalent to the Jensen's lower bound's stochastic gradient estimates.

*Proof.* The proof follows from the fact that when $q_\phi = \pi_\theta$, we have

$$g_\phi = \nabla_\theta \log \pi_\theta(c|x) \cdot \log \pi_\theta(a^*|x, c) - \nabla_\theta \log \pi_\theta(c|x)$$

Adding this gradient to $g_\theta$, a simple manipulation shows that the aggregate gradient is equivalent to the gradient of the lower bound defined in Eqn (4). □

By introducing an inference distribution $q_\phi$, ELBO is much expressive than the Jensen's lower bound and allows for a tighter approximation to the marginal log likelihood. However, this also introduces additional complexity of having to learn the approximate posterior distribution. In our applications of interest, training a posterior model of a large size can be a major computational overhead. In practice, [14] approximates the posterior via a few steps of MCMC and forgoes learning with such a distribution altogether. We take a different approach with a similar motivation: by tightening the lower bound with multiple samples, we also avoid the need for an explicit posterior.

## D.2 Reinforcement learning

We show that there is a close connection between the lower bound formulation and the expected reward (return) maximization objective in RL [65]. Concretely, we will see how the lower bound objectives are closely related to a *conditional expectation trick* that produces a RL policy gradient estimate with lower variance. First, we show that (up to a log transform) RL optimizes for the same target as the lower bound objectives, given the indicator reward.

**Lemma D.2.** (**RL optimality is equivalent to maximum likelihood optimality**) When $r(x, y) = \mathbb{1}_{\{a=a^*\}}$, the optimal policy to the RL objective is equivalent to the optimal policy of the maximum likelihood objective Eqn (2).

*Proof.* The conclusion follows from the fact that $\mathbb{E}\left[\mathbb{1}_{\{a=a^*\}}\right] = \pi_\theta(a^*|x)$. Hence the two objectives differ by a $\log$ operation and yield the same optimal solution. □

Assuming access to $n$ i.i.d. trajectories $(y_i)_{i=1}^n \sim \pi_\theta(\cdot|x)$, we start with the classic RL policy gradient with leave-one-out baseline (i.e., RLOO [54])

$$g_{\text{vanilla-pg}} = \frac{1}{n} \sum_{i=1}^n \nabla_\theta \log \pi_\theta(y_i|x) \cdot \left(\mathbb{1}_{\{a_i=a^*\}} - w_i\right), \tag{9}$$

where $w_i = \frac{1}{n-1} \sum_{j \neq i} \mathbb{1}_{\{a_j=a^*\}}$ is the leave-one-out baseline. Now, we present a new policy gradient estimate of the RL objective with guaranteed variance reduction, which is also feasible to implement with sample-based learning.

**Definition D.3** (**A variance-reduced RL policy gradient estimate**)**.** Given $n$ trajectories $(y_i)_{i=1}^n$ from a single prompt $x$, we define $g_{\text{var-reduced-pg}}$ as

$$\frac{1}{n} \sum_{i=1}^n \nabla_\theta \log \pi_\theta(c_i) \cdot (\pi_\theta(a^*|c_i) - \widetilde{w}_i) + \nabla_\theta \pi_\theta(a^*|c_i), \tag{10}$$

where $\widetilde{w}_i = \frac{1}{n-1} \sum_{j \neq i} \pi_\theta(a^*|c_j)$ is the leave-one-out baseline akin to similar constructs in the lower bound case.

We show that the variance-reduced policy gradient estimate is closely related to the classic gradient estimate via the conditional expectation trick.

**Lemma D.4.** (**Conditional expectation**) Under the same assumption as Lemma D.2 and denoting $a \sim \pi_\theta(\cdot|c)$ as the sampling process $a_i \sim \pi_\theta(\cdot|c_i)$, it holds that $g_{\text{var-reduced-pg}}$ is a conditional expectation of $g_{\text{vanilla-pg}}$

$$g_{\text{var-reduced-pg}} = \mathbb{E}_{a \sim \pi_\theta(\cdot|c)}\left[g_{\text{vanilla-pg}} \mid (c_i)_{i=1}^n\right]. \tag{11}$$

We note that without the leave-one-out baselines $\widetilde{w}_i, \widetilde{w}_i$, the conclusion Eqn (11) is straightforward as both estimates Eqn (10) and Eqn (9) become plain averages of i.i.d. terms. Now, using Lemma D.4, we immediately see that the new gradient estimate yields smaller variance.

**Theorem D.5.** (**Variance reduction**) Under the same assumption as Lemma D.2, we have guaranteed variance reduction

$$\mathbb{V}_{(y_i)_{i=1}^n \sim \pi_\theta(\cdot|x)}\left[g_{\text{var-reduced-pg}}\right] \leq \mathbb{V}_{(y_i)_{i=1}^n \sim \pi_\theta(\cdot|x)}\left[g_{\text{vanilla-pg}}\right]. \tag{12}$$

The proof is provided in Appendix E. Putting $g_{\text{var-reduced-pg}}$ from Eqn (10) and the gradient estimate of the Jensen's lower bound (Eqn (4)) side-by-side, we identify intriguing similarities. Both gradient estimates employ two terms that update either the chain-of-thought component $\pi_\theta(\cdot|x)$ or the answer component $\pi_\theta(\cdot|x, c)$, with the only subtle difference being the extra log-transform needed for obtaining the Jensen lower bound. This alludes to the fact that the lower bound gradient has intrinsic built-in variance reduction. We provide additional discussions of a few practical trade-offs in using the variance-reduced estimate $g_{\text{var-reduced-pg}}$ in Appendix E.

### D.3 Optimizing Jensen's lower bound with regularization is optimizing a special ELBO

When optimzing the lower bound objectives, we also apply the KL regularization motivated from the regularized RL formulation (Eqn (1)). Though this combination seems ad-hoc, we will see that optimizing such an hybrid objective is in fact equivalent to maximizing a special ELBO.

Incorporating the regularization into the lower bound formulation, we have an aggregate objective

$$\mathcal{L}_\theta(x, a^*) - \beta \mathbb{KL}(\pi_\theta, \pi_{\text{ref}}). \tag{13}$$

If we refine the regularization a little more: instead of the generation level regularization, we apply regularization at the chain-of-thought: $\mathbb{KL}_c(\pi_\theta, \pi_{\text{ref}}) := \mathbb{E}_{c \sim \pi_\theta(\cdot|x)} \left[ \log \frac{\pi_\theta(c|x)}{\pi_{\text{ref}}(c|x)} \right]$, then the resulting aggregate objective can be interpreted in a more coherent way, as an ELBO to a concrete generative process.

**Lemma D.6.** (**Regularized lower bound as an ELBO to a special generative process**) Assume a generative process $c \sim \pi_{\text{ref}}(\cdot|x), a \sim \pi_\theta(\cdot|x, c)$ that defines a marginal distribution $p_{\pi_\theta, \pi_{\text{ref}}}(a|x) := \sum_c \pi_{\text{ref}}(c|x) \pi_\theta(a^*|x, c)$. Then the regularized objective $\mathcal{L}_\theta(x, a^*) - \mathbb{KL}_c(\pi_\theta, \pi_{\text{ref}})$ is a lower bound to the log likelihood $\log p_{\pi_\theta, \pi_{\text{ref}}}(a|x)$.

*Proof.* Applying the same derivation as the regular ELBO, log likelihood $\log p_{\pi_\theta, \pi_{\text{ref}}}(a|x)$ is lower bounded as

$$\geq \max_\phi \mathbb{E}_{c \sim q_\phi(\cdot|x, a^*)} \left[ \log \pi_\theta(a^*|x, c) - \log \frac{q_\phi(c|x, a^*)}{\pi_{\text{ref}}(c|x)} \right]$$

$$\geq_{(a)} \mathbb{E}_{c \sim \pi_\theta(\cdot|x)} \left[ \log \pi_\theta(a^*|x, c) - \log \frac{\pi_\theta(c|x)}{\pi_{\text{ref}}(c|x)} \right]$$

$$= \mathcal{L}_\theta(x, a^*) - \mathbb{KL}_c(\pi_\theta, \pi_{\text{ref}}),$$

where inequality (a) is due to choosing $q_\phi = \pi_\theta$ and the last equality is by definition. Hence the proof is complete. $\square$

Note that the aggregate objective Eqn (13) can also be optimized via stochastic gradient ascent. We just need to add an additional term associated with the KL regularization, to the original gradient estimate to $\mathcal{L}_\theta(x, a^*)$ defined in Eqn (4). An example of such a gradient estimate usually takes the following form

$$\log \frac{\pi_\theta(c|x)}{\pi_{\text{ref}}(c|x)} \nabla_\theta \log \pi_\theta(c|x), c \sim \pi_\theta(\cdot|x).$$

Though our lower bound interpretation (Lemma D.6) is under a regularization only on the chain-of-thought, in practice, we still apply the full generation level regularization following common practice [22, 24, 23].

### D.4 Practical trade-offs compared to RL

As discussed earlier, JEPO does not require an external verifiable reward, as it can be understood as adopting the indicator reward $r(x, y) = \mathbb{1}_{\{a=a^*\}}$. In practice, this can be instantiated as a strict string match `float(answer == gt_answer)`. However, such a reward function will likely induce false negatives, as semantically equivalent generations might be vastly different strings. In practice, a more lenient match is typically applied to better balance the false negative. For example, for math problems [11, 56], usually programmtic checks are implemented to check for equivalence of two short expressions, such that e.g., `pi` and `3.1415926` might be considered equivalent.

More formally, consider a general reward function $r(x, y) = \text{match}(a, a^*)$ calculated as a binary match between $a$ and $a^*$. We can rewrite the RL objective as $\mathbb{E}[\text{match}(a, a^*)]$. In order to adapt the formulation in this work to the lower bound case, we need to work explicitly with the equivalent set $\mathcal{A} := \{a | \text{match}(a, a^*) = 1\}$. We will need to calculate quantities such as the probability $\pi_\theta(\mathcal{A}|x, c) := \sum_{a \in \mathcal{A}} \pi_\theta(a|x, c)$, which reduces to $\pi_\theta(a^*|x, c)$ in case we use exact match. Computing such probabilities is expensive since we need to enumerate all $a \in \mathcal{A}$ if inverting the match function is feasible at all. As a result, the lower bound formulation cannot be adapted to generic match function or reward function.

In summary, when a good (verifiable) reward is available (Sympy vs. string for certain math datasets, see Section 7), online RL is at an advantage. There are also cases where good rewards are not easy to come by, and JEPO is a decent default strategy. An example is where the ground truth answer takes a rather long form, in which case string match or programmatic check will produce too much false negatives, see Section 8.

## E    Variance-reduced RL gradient estimate

We provide more discussion on the variance-reduced RL gradient estimate.

### E.1    Proof of variance reduction

Recall that we denote $(y_i)_{i=1}^n$ as the set of generations and $(c_i)_{i=1}^n$ be the set of chain-of-thoughts generated from prompt $x$. We drop the dependency on prompt $x$ wherever the context is clear.

*Proof of Theorem 12.*  A direct computation shows that

$$
\begin{aligned}
\mathbb{V}_{(y_i)_{i=1}^n \sim \pi_\theta(\cdot|x)} \left[ g_{\text{vanilla-pg}} \right] =& \mathbb{E}_{(y_i)_{i=1}^n \sim \pi_\theta(\cdot|x)} \left[ g_{\text{vanilla-pg}} - g_{\text{var-reduced-pg}} + g_{\text{var-reduced-pg}} - \mathbb{E}_{(y_i)_{i=1}^n \sim \pi_\theta(\cdot|x)} \left[ g_{\text{vanilla-pg}} \right] \right] \\
=& \mathbb{E}_{(y_i)_{i=1}^n \sim \pi_\theta(\cdot|x)} \left[ \left\| g_{\text{vanilla-pg}} - g_{\text{var-reduced-pg}} \right\|^2 \right] + \mathbb{V}_{(y_i)_{i=1}^n \sim \pi_\theta(\cdot|x)} \left[ g_{\text{var-reduced-pg}} \right],
\end{aligned}
\tag{14}
$$

where the cross-term vanishes due to Eqn (11). From this, Eqn (12) follows immediately.    □

*Proof of Lemma D.4.* We begin by computing the conditional expectation $\mathbb{E}_{a \sim \pi_\theta(\cdot|c)} \left[ g_{\text{vanilla-pg}} \mid (c_i)_{i=1}^n \right]$, which yields

$$
\underbrace{\mathbb{E}_{a \sim \pi_\theta(\cdot|c)} \left[ \frac{1}{n} \sum_{i=1}^n \nabla_\theta \log \pi_\theta(y_i) \cdot \mathbb{1}_{\{a_i = a^*\}} \mid (c_i)_{i=1}^n \right]}_{\text{I}} + \underbrace{\mathbb{E}_{a \sim \pi_\theta(\cdot|c)} \left[ \frac{1}{n} \sum_{i=1}^n \nabla_\theta \log \pi_\theta(y_i) \cdot \widetilde{w}_i \right]}_{\text{II}} \cdot
\tag{15}
$$

where we use the notation $a \sim \pi_\theta(\cdot|c)$ to indicate that each answer $a_i \sim \pi_\theta(\cdot|c_i)$ is i.i.d. sampled from its corresponding chain-of-thought. Expanding the first term I, we have

$$
\begin{aligned}
\text{I} =_{(a)} & \frac{1}{n} \sum_{i=1}^n \sum_a \left( \nabla_\theta \log \pi_\theta(a|c_i) + \nabla_\theta \log \pi_\theta(c_i) \right) \cdot \mathbb{1}_{\{a = a^*\}} \cdot \pi_\theta(a|c_i) \\
=_{(b)} & \frac{1}{n} \sum_{i=1}^n \left( \nabla_\theta \pi_\theta(a^*|c_i) + \nabla_\theta \log \pi_\theta(c_i) \cdot \pi_\theta(a^*|c_i) \right),
\end{aligned}
\tag{16}
$$

where (a) is by definition of the expectation and $a \in \mathcal{A}$ denotes a dummy answer variable; (b) is due to the definition of the indcator function. Now recalling the definition of $w_i$ as leave-one-out baseline to simplify term II:

$$
\text{II} = \frac{1}{n} \sum_{i=1}^n \mathbb{E}_{a \sim \pi_\theta(\cdot|c)} \left[ \nabla_\theta \log \pi_\theta(y_i) \cdot w_i \mid (c_i)_{i=1}^n \right] = \frac{1}{n(n-1)} \sum_{i=1}^n \sum_{j \neq i} \mathbb{E}_{a \sim \pi_\theta(\cdot|c)} \left[ \nabla_\theta \log \pi_\theta(y_i) \cdot \mathbb{1}_{\{a_j = a^*\}} \mid (c_i)_{i=1}^n \right] \cdot
\tag{17}
$$

Note we can explicitly compute each summand on the right hand side of Eqn (17) as product of two conditional expectations, thanks to the fact that when $i \neq j$:

$$
\begin{aligned}
\mathbb{E}_{a \sim \pi_\theta(\cdot|c)} [\nabla_\theta \log \pi_\theta(y_i) \cdot \mathbb{1}_{\{a_j = a^*\}} \mid (c_i)_{i=1}^n] =_{(a)} & \left( \mathbb{E}_{a_i \sim \pi_\theta(\cdot|c_i)} [\nabla_\theta \log \pi_\theta(a_i|c_i)|c_i] + \nabla_\theta \log \pi_\theta(c_i) \right) \cdot \pi_\theta(a^*|c_j) \\
=_{(b)} & \nabla_\theta \log \pi_\theta(c_i) \cdot \pi_\theta(a^*|c_j),
\end{aligned}
\tag{18}
$$

where (a) is due to the definition of the indicator function; (b) is based on the zero-mean property of score functions. Plugging Eqn (18) into the right hand side of Eqn (17), we have

$$
\text{II} = \frac{1}{n} \sum_{i=1}^n \nabla_\theta \log \pi_\theta(c_i) \cdot \frac{1}{n-1} \sum_{j \neq i} \pi_\theta(a^*|c_j) = \frac{1}{n} \sum_{i=1}^n \nabla_\theta \log \pi_\theta(c_i) \cdot \widetilde{w}_i,
\tag{19}
$$

where we used the definition of $\widetilde{w}_i$ from Eqn (10). Lastly, we combine Eqn (16) and Eqn (19) and obtain

$$
\text{I} + \text{II} = \frac{1}{n} \sum_{i=1}^n \left( \nabla_\theta \pi_\theta(a^*|c_i) + \nabla_\theta \log \pi_\theta(c_i) \cdot \left( \pi_\theta(a^*|c_i) - \widetilde{w}_i \right) \right) = g_{\text{var-reduced-pg}}.
\tag{20}
$$

Thus we have concluded the proof of Lemma D.4.    □

