# OpenReview forum: "Beyond Verifiable Rewards: Scaling Reinforcement Learning in Language Models to Unverifiable Data"
_NeurIPS.cc/2025/Conference — NeurIPS 2025 poster_

### Official Review · Reviewer_aCix · 2025-06-30

**Clarity:** 3
**Significance:** 3
**Originality:** 2
**Rating:** 4
**Confidence:** 3

**Summary:**

The paper introduces JEPO (Jensen’s Evidence lower bound for Policy Optimization), an RL algorithm that dispenses with explicit, automatically checkable rewards and instead maximizes a Jensen-tightened lower bound on the log-likelihood of ground-truth answers, treating the model’s chain-of-thought as a latent variable. Compared with standard online RL from verifiable rewards (e.g., Sympy-matched answers in math datasets), JEPO requires no external reward signal, works with both verifiable and unverifiable long-form answers, and combines a REINFORCE–style term with a supervised log-probability term. Experiments on three settings—verifiable (MATH), semi-verifiable (Numina), and unverifiable (Numina-Proof)—show that single-sample JEPO matches, and its multi-sample variant often surpasses, traditional RL on evaluation metrics, while being able to exploit the entire dataset when only a subset is verifiable.

**Questions:**

see weakness

**Ethical Concerns:**

["NO or VERY MINOR ethics concerns only"]

**Limitations:**

see weakness

**Quality:**

3

**Strengths And Weaknesses:**

Strengths:

- Conceptual novelty – Re-framing chain-of-thought optimization as latent-variable modeling and applying Jensen’s inequality yields a simple, principled objective that avoids auxiliary posterior models and scales naturally to unverifiable data.

- Practical relevance – Many real-world tasks have long, hard-to-verify answers; JEPO demonstrates competitive or better performance on such datasets, highlighting genuine utility beyond synthetic short-answer benchmarks.

- Thorough empirical study – The authors test across three verification regimes (verifiable, semi-verifiable, unverifiable) and two model sizes (8 B, 70 B), include ablations on sample count, supervised-loss weight, and KL regularization, and compare with combined JEPO + RL baselines, providing a detailed performance picture.


Weaknesses:

- Evaluation of unverifiable tasks relies on proxy metrics only. For Numina-Proof (Sec. 7 & App. C.1), the authors switch to a test-set negative-log-likelihood surrogate. While JEPO shows lower NLL than RL, the paper never validates that lower NLL correlates with human-judged proof quality, leaving the headline claim (“improves reasoning on unverifiable data”) empirically under-supported. A small human study or at least an LLM-as-judge sanity check would have strengthened the evidence.
- Loose theoretical guarantees – While the Jensen lower bound is tractable, it is strictly looser than the full ELBO; the paper provides limited analysis of how the gap affects convergence or final policy optimality, leaving theoretical performance bounds unclear.
- Reward-free versus reward-rich trade-off – In verifiable settings JEPO lags behind standard RL during training and relies on KL regularization to maintain stability, suggesting efficiency drawbacks when high-quality rewards are available; the paper could discuss when one should prefer JEPO in practice.

---

> ### Author Rebuttal · Authors · 2025-07-29
>
> Thank you very much for your time and efforts in reviewing our manuscript. We will make sure to incorporate your feedback into our later revisions.
>
> Thank you very much for your constructive feedback and very positive view towards our paper!
>
> > Evaluation of unverifiable tasks relies on proxy metrics only. For Numina-Proof (Sec. 7 & App. C.1), the authors switch to a test-set negative-log-likelihood surrogate. While JEPO shows lower NLL than RL, the paper never validates that lower NLL correlates with human-judged proof quality, leaving the headline claim (“improves reasoning on unverifiable data”) empirically under-supported. A small human study or at least an LLM-as-judge sanity check would have strengthened the evidence.
>
> Thank you for raising the suggestion. Indeed we agree that the result can be strengthened further by correlating NLL with human-judged proof quality. We did not yet have the capacity to carry out the human study by the time of the publication, so tried to resort to test set likelihood as a proxy. With negative results of using automated proof judge [1], we feel that LLM-as-judge for the proof data might not deliver very reliable eval results either.
>
> We did have some manual qualitative assessments of the trained model: while the baseline outputs a proof directly, JEPO trained model outputs a high-level proof strategy followed by the detailed proof. This might corroborate with why the test set NLL is better, due to the learned CoT strategy before the detailed proof.
>
> Despite the limitations, we would still like to argue that the comparison is apple-to-apple, with a reasonable baseline.
>
> [1] Petrov et al, 2025, Proof or Bluff: Evaluating LLMs on 2025 USA Math Olympiad
>
> > Loose theoretical guarantees – While the Jensen lower bound is tractable, it is strictly looser than the full ELBO; the paper provides limited analysis of how the gap affects convergence or final policy optimality, leaving theoretical performance bounds unclear.
>
> We acknowledge the theoretical looseness with the bound. With theory, we can seek to close the bound by increasing $n$, but fundamentally this is what we could do with such a methodological design.
>
> We have extensively assessed the empirical impact of such a theoretical loose bound in experiments. Fig 1 showed that despite with a lower bound objective, we can optimize the target reward function; Fig 2 showed that increasing $n$ to $n=4$ correlates with the reward objective better than $n=1$, which corroborates with the performance ablations of Figure 2 across different $n$s.
>
> The high-level message is that while the lower bound objective might be a loose objective, its looseness can be mitigated largely in practice by choosing a reasonable value of $n$. In our ablations, $n=4$ seems to suffice for the math related domains. And we leave to practitioners to understand the impact of this hyper-parameter in their specific settings.
>
> > Reward-free versus reward-rich trade-off – In verifiable settings JEPO lags behind standard RL during training and relies on KL regularization to maintain stability, suggesting efficiency drawbacks when high-quality rewards are available; the paper could discuss when one should prefer JEPO in practice.
>
> Indeed JEPO relies on KL regularization slightly more than RLVR in our ablations, but we feel that’s not a strong constraint. More broadly, we find that on domains where data is short-form and the reward is quite accurate, RLVR retains its advantage compared to JEPO (as we have shown in Fig 1). That said, JEPO is still quite competitive in this regime.
>
> Yet in semi-verifiable or non-verifiable domains, JEPO has an advantage due to its more flexible learning objective (without a reliance on good reward). In practical situations where semi-verifiable or non-verifiable data are abundant, JEPO might be worth a try for practitioners.

---

### Official Review · Reviewer_roE2 · 2025-07-01

**Clarity:** 3
**Significance:** 2
**Originality:** 3
**Rating:** 4
**Confidence:** 3

**Summary:**

The authors propose a policy optimization algorithm that targets reinforcement learning with unverifiable rewards. The algorithm, namely Jensen's Evidence lower bound for Policy Optimization (JEPO), leverages Jensen's evidence lower bound, which is based on a latent variable perspective of the LLM's chain-of-thought. The algorithm itself also presents a much simpler objective and makes RL more scalable. The authors provide both theoretical justification and empricial evidence to demonstrate the effectiveness of the algorithm on MATH and Numina.

**Questions:**

1. In the introduction (line 28), the authors define data as unverifiable if the ground truth answer to the problem cannot be verified with reasonable automated effort. If I understand correctly, this definition still assumes the availability of the answer. Is that correct? If so, do we also care about the quality and the uniqueness of the answer?
2. In Section 5, the authors indicate the difficulty of using LLM-as-a-Judge as a way to tackle unverifiable data. Do the authors have concrete examples of this, even when the (unverfiable) ground truth answer is available?

**Ethical Concerns:**

["NO or VERY MINOR ethics concerns only"]

**Final Justification:**

I decided to keep my original score of this paper after the rebuttal. My major concern in the original review is that the paper only demonstrate limited evaluation in in-distribution/domain settings, which is rarely the case the real world.

Specifically, the authors have only demonstrated evaluation on the MATH and NuminaMath test sets. While this approach is ideal for experiments, questions about whether it truly transfers to more challenging and mainstream benchmarks like the AIME remain.

The authors have provided sufficient clarifications in the rebuttal; however, they have not included experimental results to address my concerns about the practicality of the method.

**Limitations:**

yes

**Quality:**

3

**Strengths And Weaknesses:**

Strengths:

1. The proposed algorithm is shown to be espeically effective in the scenarios of math problems with hard-to- or non-verifiable ground truth.
2. The core policy optimization algorithm is built upon Jensen's lower bound and is theoretically sound.
3. The paper is clearly written and well-structured. It follows standard notation and is easy to understand.

Weaknesses:

1. The theoretical section of the paper is robust, but the experiments presented are somewhat lacking.
    - The main experiments are based on MATH and Numina only, which only assesses in-distribution problems. This is important as we do not always have in-distribution data for training.
    - In the Numina benchmark, roughly 40% of training examples cannot be scored by SymPy. As the authors note, the baseline RL agent therefore optimizes only a subset of the data, and JEPO is evaluated under conditions where a large fraction of examples have no ground-truth verification. This leaves open whether the method scales when the majority of data are truly unverifiable, or when tasks fall outside the narrow domain of mathematical proofs.
    - In the core MATH benchmark the authors state "We do not compare with other baselines developed in prior work." This leaves open the possibility that apparent gains might come from experimental setup rather than the proposed JEPO algorithm.
2. Some minor points:
    - For MATH evaluation, it might be more ideal to consider MATH-500, which are less likely to be contaminated.
    - SymPy is known to have many parsing issues specifically for MATH [2], so the reward may not be entirely accurate either.
    - The authors use Llama 3 70B as the LLM-as-a-Judge for evaluation. The authors also mention the drawback of using LLM-as-a-Judge for domain-specific and long-form data. While Llama 3 70B is a strong model, it may be a less competitive option compared to some of the leading reasoning models available today, especially on this particular task of verifying proofs.
    - For the semi- and unverifiable splits, success is measured by a composite score that falls back on majority voting from Llama 3 70B when automatic SymPy checks fail. Because of the above reasoning, it makes it hard to know whether gains are real reasoning improvements or artifacts of that judge's potentially deviated preferences.

References

[1] Lightman, H., Kosaraju, V., Burda, Y., Edwards, H., Baker, B., Lee, T., ... & Cobbe, K. (2023, May). Let's verify step by step. In The Twelfth International Conference on Learning Representations.

[2] Kydlicek, H., Lozovskaya, A., Habib, N., & Fourrier, C. (2025, February 14). Fixing Open LLM Leaderboard with Math‑Verify. Hugging Face Blog.

---

> ### Author Rebuttal · Authors · 2025-07-29
>
> Thank you very much for your time and efforts in reviewing our manuscript. We will make sure to incorporate your feedback into our later revisions.
>
> Thank you very much for your constructive feedback and very positive view towards our paper!
>
> > The main experiments are based on MATH and Numina only, which only assesses in-distribution problems. This is important as we do not always have in-distribution data for training.
>
> We acknowledge this is a limitation of this work, that the training and testing sets are mostly in distribution. That said, we think this setting offers a more narrow yet more controlled investigation into performance gains, and maybe rule away slightly confounding factors that impact how out of distribution generalization works.
>
> We would hope that investigations into out of distribution generalization can be pursued on its own right in another follow up work, or in industry labs where such considerations are more practically meaningful. Nevertheless, we hope this work offers some limited insights.
>
> > In the Numina benchmark, roughly 40% of training examples cannot be scored by SymPy. As the authors note, the baseline RL agent therefore optimizes only a subset of the data, and JEPO is evaluated under conditions where a large fraction of examples have no ground-truth verification. This leaves open whether the method scales when the majority of data are truly unverifiable, or when tasks fall outside the narrow domain of mathematical proofs.
>
> Thank you for the comments on this. We have indeed manually checked that among the *non-proof* Numina dataset, there is *still* a significant portion (about 40%) of the examples where the answers are in between short-form (e.g., a single number of a single expression) and very long-form (e.g., a full proof). These answers are usually a bit elaborate and cannot be checked by the Sympy scorer we implemented for the experiments.
>
> In Sec 7, when training on the semi-verifiable Numina dataset, we actually excluded the proof Numina sub-set, and only trained on the non-proof section of the dataset. If we understand your comment correctly, this would mean that our approach can scale to cases where a good portion of the data is unverifiable (40%), though not necessarily “majority of the data”.
> We think it is a meaningful ablation to understand whether JEPO works in the presence of a majority of unverifiable data. That said, in practice, data collection works jointly with algorithmic design, and if the in-the-wild Numina dataset contains 40% unverifiable data, it might be less interesting for a case where most data is unverifiable. Would the reviewer has in mind some dataset like this?
>
> > In the core MATH benchmark the authors state "We do not compare with other baselines developed in prior work." This leaves open the possibility that apparent gains might come from experimental setup rather than the proposed JEPO algorithm.
>
> Thank you for pointing this out. Maybe more precisely, what we meant is that: (1) We compare with a RL algorithmic baseline in a way that minimizes the discrepancy between experimental setups; (2) There are other RL algorithmic variants, with changes complementary to our JEPO algorithm.
>
> By (1), we practically mean that the only difference between JEPO vs. RL baseline, is the loss function. They share other settings such as hyper-parameters, training-generation loop, leave-one-out control variates construct etc. This rules out confounding factors as the reviewer worries about as much as possible, leading us to conclude more confidently that the improvements or trade-offs are due to loss functions.
>
> It is also important to note that since we do on-policy training, our algorithm recovers state-of-the-art algorithms like GRPO (when GRPO is on-policy, its importance ratios).
>
> By (2), we mean that there are changes such as different ways of doing group aggregation (e.g., difference between Dr. GRPO vs. GRPO), which are complementary to the designs of our algorithm. That is, in principle, such algorithmic changes can be combined with our approach, for which we find an ablation offers less insight.
>
> > For MATH evaluation, it might be more ideal to consider MATH-500, which are less likely to be contaminated.
>
> Thank you for the suggestion on this. We can try to carry out additional evaluation on this if the reviewer finds it useful.
>
> > SymPy is known to have many parsing issues specifically for MATH [2], so the reward may not be entirely accurate either.
>
> Thank you very much for the suggestion. We have some additional fixes inside the “Sympy” parser that actually removes some of the false positives and negatives suggested in the paper you shared, but still there might be some inaccuracies remaining. We will make sure to discuss this point in the revision.
>
> > The authors use Llama 3 70B as the LLM-as-a-Judge for evaluation. The authors also mention the drawback of using LLM-as-a-Judge for domain-specific and long-form data. While Llama 3 70B is a strong model, it may be a less competitive option compared to some of the leading reasoning models available today, especially on this particular task of verifying proofs.
>
> We agree with the reviewer’s comment on this. We used LLM-as-judge mainly to assess semi-long-form data, so data in Numina which is not as long-form as proof, but is more elaborate than a single mathematical expression that can be verified by Sympy. We find LLM-as-judge to work reasonably well in this case, even with Llama 70B, though indeed more powerful models can potentially provide more accurate assessments.
>
> To provide some calibration results using LLM-as-judge for the Llama 70B model: with manual checks, we find that Sympy correlates with human 80% while LLM-as-judge can improve it to 90%. But potentially more competitive models can offer even higher correlation with some tweaks or tuning, which we leave to further investigations.
>
> We want to emphasize that we did not use LLM-as-judge to verify proof in the paper. The main reason is due to the negative result shared in [1], where the authors shared discrepancies between LLM-as-judge from frontier models and human annotators. Though the Numina proof dataset might be distributionally quite different from the Math Olympiad as studied in [1], we feel less confident in using LLM-as-judge score for proof verifications.
>
> [1] Petrov et al, 2025, Proof or Bluff: Evaluating LLMs on 2025 USA Math Olympiad
>
> > For the semi- and unverifiable splits, success is measured by a composite score that falls back on majority voting from Llama 3 70B when automatic SymPy checks fail. Because of the above reasoning, it makes it hard to know whether gains are real reasoning improvements or artifacts of that judge's potentially deviated preferences.
>
> We acknowledge that LLM-as-judge might potentially have its deviated preferences, e.g., to certain formatting. In our early investigation of LLM-as-judge, we do find that the judge model can alleviate more false positives than the false positives it introduces compared to Sympy checker. As evidence of this, we have found that training against LLM-as-judge signals helps improve evaluation performance to a level slightly better than training against just limited Sympy signals.
>
> Our early manual check of LLM-as-judge found that it is quite accurate for assessing medium-form data in the semi-verifiable dataset (note that they are generally not as long as full proof, but rather elaborate short-form answers). We can provide a more accurate evaluation of LLM-as-judge against human annotations in the revision, in case the reviewer finds it useful.
>
> > In the introduction (line 28), the authors define data as unverifiable if the ground truth answer to the problem cannot be verified with reasonable automated effort. If I understand correctly, this definition still assumes the availability of the answer. Is that correct? If so, do we also care about the quality and the uniqueness of the answer?
>
> That is correct, we still assume access to a ground truth answer.
>
> Regarding quality: we think it is definitely important to guarantee that the answers are of high quality. Indeed, this will directly feed into how well the model learns from the training set and impacts its ability to generalize. We feel this argument applies for both regular RLVR and JEPO.
>
> Regarding uniqueness: we believe this is quite a subtle question and whose impact might warrant more careful investigations. In RLVR, the verifier or reward function judges if an answer is equivalent to the ground truth answer, and so it kind of assumes that the ground truth might be non-unique (otherwise we’d use a string match reward). It might be interesting to investigate further how the diversity or uniqueness of the answers, impact RLVR and JEPO in more practical scenarios.
>
> > In Section 5, the authors indicate the difficulty of using LLM-as-a-Judge as a way to tackle unverifiable data. Do the authors have concrete examples of this, even when the (unverfiable) ground truth answer is available?
>
> We meant to indicate that LLM-as-judge can be used for assessing the equivalence of model generated answer and an available *ground truth answer*, hence replacing the role of a Sympy checker in more unverifiable domains. That is, our discussion always assumes access to a ground truth reward.
>
> In case a ground truth reward is not available, it feels more akin to the conventional RLHF settings where a reward model trained from human preference or a judge model directly assesses the quality of responses. Along this line, [1] is an example where the LLM is directly prompted as a judge to deliver an evaluation score (without access to ground truth).
>
> [1] Lee et al, 2024, RLAIF vs. RLHF: Scaling Reinforcement Learning from Human Feedback with AI Feedback

---

> > ### Comment · Reviewer_roE2 · 2025-08-06
> >
> > I appreciate the authors' efforts to clarify both the weaknesses and the questions raised. Your response to the questions has addressed most of my confusion.
> >
> > My main concern is the practicality of the proposed method and the limited evaluation. If the authors can demonstrate the effectiveness of the proposed approach in similar or different domains (i.e., not only in iid) in the future, this would make for a much stronger paper and could have implications for the future direction of the field. However, I understand the resource constraints in academia. Therefore, I will maintain my current positive rating.

---

> > > ### Author Response · Authors · 2025-08-06
> > > **Thank you very much for youre reply!**
> > >
> > > Thank you for your time in engaging with our rebuttal, we appreciate your time!
> > >
> > > > If the authors can demonstrate the effectiveness of the proposed approach in similar or different domains (i.e., not only in iid) in the future, this would make for a much stronger paper and could have implications for the future direction of the field.
> > >
> > > Thank you for raising this point. The train/test iid setting is a canonical ML setting for LLM benchmarking as well. Though we understand this assumption might not hold in practice (i.e., test set can be out of distribution from training), but this is also the best we can do in a more controlled setting, and a setting where reasonable insights can be gathered.
> > >
> > > Does this echo with the reviewer's thoughts? Though we argue that this particular context is sufficiently expressive to test the new approach, we are curious about the setup that the reviewer has in mind.
> > >
> > > Many thanks please let us know if you have additional questions or thoughts.

---

### Official Review · Reviewer_kgcS · 2025-07-02

**Clarity:** 3
**Significance:** 2
**Originality:** 2
**Rating:** 3
**Confidence:** 4

**Summary:**

The paper considers the problem setting of unverifiable data, where it's difficult to judge the ground truth correctness of arbitrary answers (e.g., long-form non-formalized mathematical proofs). Noting that RL is difficult to use in such situations, the authors propose JEPO, which maximizes a lower bound to the log likelihood of a (assumed available) ground truth answer. Critically, this differs from directly supervised finetuning on the answer because the authors specifically consider the setting with chain-of-thought, so the log likelihood involves an expectation over chain-of-thoughts. The authors also extend this to a multi-sample bound. The authors evaluate on math datasets, ranging from settings where the answer is verifiable to unverifiable, comparing JEPO with varying levels of samples against an RL baseline.

**Questions:**

See questions above as well. Below are some more clarifying questions.

Line 87 - regarding the motivation, is this really what we want? Note that this does not directly enforce correct chains of thoughts. The objective could be satisfied by the LM always outputting the right answer, for any arbitrary chain of thought, including wrong chains of thoughts. I understand that it's hard to enforce correctness on the chain of thought without something more expensive like process reward models, but I'd be interested in seeing more analysis around whether this is a problem, and if not, why not.

$g_2^{(n)}$ is itself a lower bound right? Seems worth discussing and investigating more. Also, how does this compare to the unbiased $g_2$ term in Eq. 4?

I only took a brief look at VR-CLI but it seems to me more different from your work than your writing suggests? Maybe worth more explanation.

Lines 334-335 - where in Section 6 do you show the different $\beta_{sup}$?

Overall, I am on the fence about this work. The paper explores interesting directions and there are many good ideas in it. But unfortunately I think the evaluations are not strong, and the weaknesses above are a bit too much, leading me to borderline reject instead of borderline accept. But I would consider increasing my score if the authors convincingly address my points above.

**Ethical Concerns:**

["NO or VERY MINOR ethics concerns only"]

**Final Justification:**

While the discussion with the authors has improved my evaluation of the paper slightly, I am not sure it's enough to push me from borderline reject to borderline accept. I am still very much on the fence about this work (as it stands). If the AC/meta-reviewer wants to accept this paper, I would consider that a reasonable decision (though still personally weakly disagree with that).

**Limitations:**

There is much more the authors could discuss on limitations (e.g., my points above).

The authors claim that "Our work is algorithmic and theoretical in nature and does not produce immediate societal impacts", but I disagree with this. If there is any chance the work may have impact on e.g., improving the mathematical ability of frontier models, this could have significant societal impacts. Any capability advances come with possible harm in the hands of bad actors. While I agree that this work in particular is unlikely to have negative (or positive) societal impacts, I wouldn't say that chance is 0.

**Paper Formatting Concerns:**

No major formatting concerns.

**Quality:**

2

**Strengths And Weaknesses:**

Strengths:

Problem setting is interesting and worth exploring; improvements could lead to advances in important areas such as proof generation.

Overall ideas are generally good, and authors consider and explore several worthwhile directions.

Core theory seems mostly correct and reasonable to me.

Paper is not too hard to understand.

Weaknesses:

Line 70 - should be "a desirable".

Lines 78-80 - not grammatically correct. Maybe an easy fix for this is "getting" -> "we get".

Eq 4 is not strictly correct; the expectation is omitted.

Line 122 - should be "introduces".

Line 127 - I recommend "yet a tighter" -> "yet tighter" and removing the comma "alternative, is" -> "alternative is".

It is OK to refer to previous papers for derivations (as is done several times throughout), but it would be better if at least the Appendix includes more details, rather than a single reference, even if it is as simple as copying math and changing the notation to match the rest of the paper. The less time and effort the reader has to expend to understand the work, the better.

Line 133 - what are these regularity conditions? Should be stated, at least in the Appendix.

First term of Eq. 7 I believe is missing a 1/n.

Line 146 - "practice ." has an extra space.

Line 160 reads a bit awkwardly, maybe "their update only partially optimizes for the..." would be better.

Contrary to the checklist point 7, I don't see any error bars, confidence intervals or tests of statistical significance for the experiments. Are the authors referring to the lightly shaded dots (which I presume are from a separate seed)? Those are not error bars.

I would suggest moving the SFT comparison from the Appendix to the main paper (as an additional line on the existing plots). Also, instead of just saying in line 235 that there are a bunch of ablations in the Appendix, it would be good to briefly outline what those ablations are (e.g., comparison vs. SFT, regularization, model size, etc.).

Line 203 - hyperparameters like batch size being constant is good for fair comparison, but not all hyperparameters should be constant. Things like learning rate should be tuned (at least using a coarse search) for each method.

Lines 206-208 - I don't think this is good reasoning for not comparing against other work. Different RL methods can have dramatically different results, and different RL algorithms have much more than "low-level implementation differences". Even if there are only "low-level implementation differences", these can still have material impacts on performance. What RL algorithm is used in the paper?

Lines 227-230 - on Figure 1, the RL frontier is clearly to the left of the JEPO frontier, except for very high KL divs. For very high KL divs, you might be running into the limits of model capacity, which may explain why performance is similar. In any case, the point of building these plots and frontiers is to look at the whole frontier, and overall the RL frontier looks much better.

Figure 2 - it's interesting and good to verify but perhaps unsurprising that RL, which is based on a sympy scorer and not exact match, would have lower NLL than your objective which includes in it an essentially SFT component on the (train set) answers. Later on I see that this is discussed, but the organization is kind of awkward - the figure appears way before the discussion, and the discussion of Figure 2b comes after discussing Figure 3. I get there are space constraints and this is space efficient, but it's probably worth trying to reorganize things or maybe even use in-line figures so the figures and paper content flow better.

Lines 260-261 - it seems important to me to quantify how many false negatives are mitigated. If checking the entire dataset is too much work, you could at least randomly sample a subset and say what proportion of false negatives were corrected. Another thing that would be important to check is how many false positives are added. LLM-as-judge might also incorrectly think some good-looking or close-enough answers are correct. The LLM-as-judge evaluation might possibly be biased in favor of JEPO over RL given that the authors' previous NLL results imply that RL is finding different, possibly less standard solutions - these seem like the kinds of solutions an LLM-as-judge would be more likely to miss. Furthermore, if JEPO outputs things that look more like the test set solutions, but are subtly wrong, the LLM-as-judge may catch fewer wrong things from JEPO relative to RL. To check this isn't a concern, I think it's pretty important to have some quantification of the false positive and false negative rates across RL answers and JEPO answers on some subset of data.

Line 288 - seems like it should be "with test set" instead of "with set".

Lines 335-338 - I have a simpler alternate explanation. Here is evaluating NLL only, whereas previous evaluation was based on sympy and LLM-as-judge. It's not really surprising that NLL is more impacted by the supervised learning component than metrics like sympy scoring which are not as directly related.

There should probably be a conclusion section, even if a short one.

(Not sure where to put this, so putting here, though only sort of a weakness): originality of the paper I'm saying is fair; there's novelty in the application of ideas to this problem setting, but I won't give a higher score since I think it's close enough conceptually to existing work on e.g. ELBO, variational inference, IWAE, etc. (this is also a reason why a lot of the proofs in the paper are simply references to existing works)

---

> ### Author Rebuttal · Authors · 2025-07-29
>
> Thank you very much for your time and efforts in reviewing our manuscript. We will make sure to incorporate your feedback into our later revisions.
>
> > Line 70… Line 160 reads a bit awkwardly...
>
> Thank you very much for the extremely detailed feedback on typos and grammar improvements, we will make sure to revise in our next manuscript.
>
> > Contrary to the checklist point 7, I don't see any error bars..
>
> Indeed, lightly shaded dots are results from multiple runs with the same configuration. In our own experience, we have found in some cases, it is a bit more informative to display the full distribution over different runs rather than standard errors or confidence intervals. We’d be happy to provide additional information such as error bars and confidence intervals if the reviewer finds them useful.
>
> > I would suggest moving the SFT comparison from the Appendix to the main paper...
>
> We are definitely happy to move the SFT ablation discussion and more details into the main paper as the reviewer suggests. The current formatting in the paper is primarily due to space limitations, but if we are allowed an extra page in camera ready, additional results can be presented in the main paper.
>
> > Line 203 - Things like learning rate should be tuned (at least using a coarse search) for each method.
>
> It is true that hyper-parameters like learning rate can be ablated in this case. In our early investigation, we have found the evaluation performance to be reasonably robust against learning rate variations, though the training curve gets impacted slightly (e.g., within a stable interval of learning rate, increasing the learning rate leads to faster growth in training reward across the board). We can provide additional information in the revision.
>
> Besides learning rate, we have indeed included other hyper-parameters for comparison, which we find to be central to RLVR methods in general, such as the number of samples per prompt $n$ and the regularization coefficient $\beta$.
>
> > Lines 206-208 - I don't think this is good reasoning for not comparing against other work... What RL algorithm is used in the paper?
>
> We agree that some technical details can make a material difference in practice. Maybe a more precise way to state the rationale here, is that some algorithmic variations are orthogonal to our algorithmic proposals in this work, and can be readily combined with our method in a complementary way. For such cases, we argue it might offer less value to carry out a comparison.
>
> The algorithm we compare against is a simple version of RLOO (Reinforce with leave-one-out [1]. We argue it is the backbone for most existing RL algorithms such as GRPO [2] and its variants. Indeed, when on-policy (which is the experimental setup we adopt), GRPO and its variants will fall back to RLOO. RLOO also shares the most algorithmic structure with our method (e.g., leave-one-out control variate), which we find a comparison might be the most insightful.
>
> [1] Ahmadian et al, 2024, Back to Basics: Revisiting REINFORCE...
>
> [2] Shao et al, Pushing the limits of mathematical reasoning...
>
> > Lines 227-230 - on Figure 1, the RL frontier is clearly to the left of the JEPO frontier, except for very high KL divs..
>
> We acknowledge that RL traces out a better KL-reward frontier in this case. But we have emphasized a few points in the paper: (1) It is the training plot where RL is better, and on evaluation plot (mid plot), JEPO is as competitive; (2) RL is at an advantage here because it has access to the reward function to plot against too, unlike JEPO, which does not assume access to the reward.
>
> The key message is that Figure 1 shows a case where RLVR should work the best (i.e., short-form data with a good verifiable reward). Section 7-8 shows the results on semi-verifiable and unverifiable domains, where JEPO methods shine. This is a practical trade-off we seek to highlight in the paper.
>
> > Figure 2 - it's interesting and good to verify but perhaps unsurprising that RL ...
>
> Thank you for finding the results interesting. Indeed, the results are meant to display a potential misalignment between likelihood based approach and Sympy based approach. It is indeed a limitation from the space constraints that causes our sub-optimal arrangement of texts and figures in the paper, we will make improvements in the revision.
>
> > Lines 260-261...  I think it's pretty important to have some quantification of the false positive and false negative rates across RL answers and JEPO answers on some subset of data.
>
> We acknowledge that LLM-as-judge might have its limitations. In our early investigation of LLM-as-judge, we did find that the judge model can alleviate more false positives than the false positives it introduces. As evidence of this, we have found that training against LLM-as-judge signals helps improve evaluation performance to a level slightly better than training against just limited Sympy signals.
>
> With some manual check, we find that Sympy correlates with human 80% of the time while judge correlates with 90%. We find that the judge’s improvement on Sympy’s false negative overwhelms its potential loss on false positive, resulting in an accuracy that’s higher than just using judge. Hence we use a combined score for eval. We are happy to provide more detailed LLM-as-judge calibration results in revision.
>
> > Lines 335-338 - I have a simpler alternate explanation...
>
> We agree with the reviewer’s interpretation, this is indeed the conclusion that we seek to find more empirical evidence for, since it is not obvious by default.
>
> > originality of the paper I'm saying is fair
>
> Despite the strong connections to prior work such as ELBO, VAE, IWAE, we believe that there are many technical contributions on top which distinguish our work: (1) JEPO is the first implementation of such ideas “as they are developed mathematically”. Though prior work borrows high level ideas, the implementations turn out to be very much similar to RLVR, see discussion in 170-175; (2) JEPO has unique technical contributions specific to LLM, instrumental to actually making the system “work”, e.g., how to resolve certain technical constraints specific to LLM. Indeed, it is not trivial to apply VAE ideas to LLM out of the box.
>
> While JEPO is not groundbreaking in its mathematical foundation (which is the same as VAE), we argue that lots of important work in space is not either. Rather, we feel technical solidity should take more priority over novelty in the space of ideas.
>
> > Line 87 - regarding the motivation, is this really what we want? ... but I'd be interested in seeing more analysis around whether this is a problem.
>
> We believe JEPO does not resolve the fundamental CoT correctness issue arising in conventional outcome based RLVR scenarios. Indeed as the reviewer suggested, we cannot do better unless a process supervision signal is available.
>
> Our speculation is that by leveraging potentially more unverifiable and semi-verifiable data, JEPO can benefit from a larger effective training set, compared to RLVR with a more rigorous reward function. In doing so, the model might overfit less to the final ground truth answer, and hence reducing the effect of “bad CoTs leading up to correct answer”. We do not have a systemic evaluation of this, rather we only evaluate the test set performance as a proxy metric.
>
> >  is itself a lower bound right? Seems worth discussing and investigating more. Also, how does this compare to the unbiased term in Eq. 4?
>
> Note that $g_2^{(n)}$ is itself not a lower bound, but rather a partial gradient to the $n$-sample lower bound (see Eqn 7).
>
> Eqn 4 shows $g_2$ for a single sample lower bound, which can be seen as a special case to the $n$-sample lower bound with $n=1$.
>
> Comparing the two: these are gradients to different lower bounds. In general, the bound is tighter as $n$ increases (see discussion near line 131), but the variance of the gradient might increase as well, causing a trade-off in principle. Figure 2 shows the empirical comparison that indicates that increasing $n$ in general seems to be helpful, though there is also the practical constraint of the algorithm being more expensive in $n$ due to generations. We will highlight more such discussions in the revision.
>
> > I only took a brief look at VR-CLI but it seems to me more different from your work than your writing suggests?
>
> We leaned on the more conservative side and discussed some similarities between VR-CLI and our approach. To be fair, VR-CLI is on a high-level similar to using likelihood as a reward, which bears intuitively resemblance to the first part of the gradient $g_1, g_1^{(n)}$ in this case. Yet, many technical details might differ: VR-CLI is not theoretically grounded to improve the likelihood objective, as we have discussed in this work with the lower bound formulation, which requires transformation of individual log likelihood.
>
> > Lines 334-335 - where in Section 6 do you show the different
>
> This was a discussion of early investigations we have: we found that when $\beta_\text{sup}$ is large, it tends to produce worse results with short-form answer dataset like MATH. We speculate this is because the force of SFT loss is too large with large values of $\beta_\text{sup}$, and causes the training to overfit more (after all, it’s simple to predict just a few tokens worth of answers, compared to long-form answers). We will provide more detailed results in the revision.
>
> > There is much more the authors could discuss on limitations.
>
> We are happy to discuss the limitations as the reviewer pointed out, given more space in the camera ready revisions.
>
> > The authors claim that "Our work is algorithmic and theoretical in nature and does not produce immediate societal impacts"
>
> We agree that if the algorithmic improvements reach certain points, it can generate sizable social impact. We are happy to provide more discussions on this in the checklist and in the revision.

---

> ### Comment · Reviewer_kgcS · 2025-08-01
>
> Thanks for your response!
>
> Quotes below abbreviated for character limit.
>
> > Indeed, lightly shaded dots are results from multiple runs with the same configuration. [...] We’d be happy to provide additional information such as error bars and confidence intervals if the reviewer finds them useful.
>
> Yes, please provide error bars/confidence intervals, with at least 3 seeds (ideally 5 or 10).
>
> > It is true that hyper-parameters like learning rate can be ablated in this case. In our early investigation, we have found the evaluation performance to be reasonably robust against learning rate variations, though the training curve gets impacted slightly (e.g., within a stable interval of learning rate, increasing the learning rate leads to faster growth in training reward across the board). We can provide additional information in the revision.
>
> > Besides learning rate, we have indeed included other hyper-parameters for comparison, which we find to be central to RLVR methods in general, such as the number of samples per prompt $n$ and the regularization coefficient $\beta$.
>
> Yes, please provide additional information, including which hyperparameters you compared and tuned, which were most important, what ranges you roughly considered, how you tuned, etc.
>
> > We agree that some technical details can make a material difference in practice. Maybe a more precise way to state the rationale here, is that some algorithmic variations are orthogonal to our algorithmic proposals in this work, and can be readily combined with our method in a complementary way. For such cases, we argue it might offer less value to carry out a comparison.
>
> > The algorithm we compare against is a simple version of RLOO (Reinforce with leave-one-out [1]. [...]
>
> Thanks for the clarification. The paper could be reworded to better reflect this.
>
> > We acknowledge that RL traces out a better KL-reward frontier in this case. But we have emphasized a few points in the paper: (1) It is the training plot where RL is better, and on evaluation plot (mid plot), JEPO is as competitive; (2) RL is at an advantage here because it has access to the reward function to plot against too, unlike JEPO, which does not assume access to the reward.
>
> Sorry, I still strongly disagree with JEPO being competitive even on the evaluation plot. I reiterate my original comment which was meant for both the train and evaluation plots: "the RL frontier is clearly to the left of the JEPO frontier, except for very high KL divs. For very high KL divs, you might be running into the limits of model capacity, which may explain why performance is similar. **In any case, the point of building these plots and frontiers is to look at the whole frontier, and overall the RL frontier looks much better.**"
>
> > The key message is that Figure 1 shows a case where RLVR should work the best (i.e., short-form data with a good verifiable reward). Section 7-8 shows the results on semi-verifiable and unverifiable domains, where JEPO methods shine. This is a practical trade-off we seek to highlight in the paper.
>
> Maybe (I'm not entirely sure about this) it might even be better to just cut out Figure 1 entirely (or leave it to the Appendix, as a sort of ablation/study just for understanding purposes), and instead do more experiments in the settings where JEPO shine, to further illustrate the usefulness of JEPO over existing RL methods. That is, you generally don't convince others by showing a setting where your method is expected to be disadvantaged, and then demonstrating that you are disadvantaged (even if less so than one might initially expect). Your efforts are probably better spent focusing on demonstrating where your method is advantaged (and justifying why those situations where your method is advantageous are of practical relevance/importance).
>
> > [...] We are happy to provide more detailed LLM-as-judge calibration results in revision.
>
> That would be great.
>
> > We agree with the reviewer’s interpretation, this is indeed the conclusion that we seek to find more empirical evidence for, since it is not obvious by default.
>
> If you agree, you should edit the paper to reflect this.
>
> > Despite the strong connections to prior work such as ELBO, VAE, IWAE, [...] Indeed, it is not trivial to apply VAE ideas to LLM out of the box.
>
> > While JEPO is not groundbreaking in its mathematical foundation (which is the same as VAE), we argue that lots of important work in space is not either. Rather, we feel technical solidity should take more priority over novelty in the space of ideas.
>
> Yes, that's why I'm saying it's fair, not bad.
>
> > [...] We will highlight more such discussions in the revision.
>
> Great, thanks for the additional clarification.
>
> > [...] We will provide more detailed results in the revision.
>
> Great, thanks.

---

> > ### Author Response · Authors · 2025-08-03
> > **Thank you for your reply!**
> >
> > Many thanks to the reviewer for replying to our rebuttal response, we appreciate your time!
> >
> > Besides the revisions we will do, a few more follow-up responses to the reviewer's replies.
> >
> > > Sorry, I still strongly disagree with JEPO being competitive even on the evaluation plot. I reiterate my original comment which was meant for both the train and evaluation plots: "the RL frontier is clearly to the left of the JEPO frontier, except for very high KL divs. For very high KL divs, you might be running into the limits of model capacity, which may explain why performance is similar. In any case, the point of building these plots and frontiers is to look at the whole frontier, and overall the RL frontier looks much better."
> >
> > We agree that in this setting RLVR obtains the best performance, and maybe we can attribute this to the fact that RLVR has access to a fairly accurate reward function, which is used for plotting both training and testing evals for Figure 1 (a)-(b). In the meantime, JEPO by design does not leverage such information and is at a disadvantage. We think it is not unfair to claim that RLVR obtains the better overall frontier here.
> >
> > By "competitive", we meant to say that JEPO is still able to obtain most of the performance improvements achieved by RLVR, so is a decent competitor to RLVR even in this setting where we expect RLVR to shine the most.
> >
> > > Maybe (I'm not entirely sure about this) it might even be better to just cut out Figure 1 entirely (or leave it to the Appendix, as a sort of ablation/study just for understanding purposes), and instead do more experiments in the settings where JEPO shine, to further illustrate the usefulness of JEPO over existing RL methods. That is, you generally don't convince others by showing a setting where your method is expected to be disadvantaged, and then demonstrating that you are disadvantaged (even if less so than one might initially expect).
> >
> > We understand that the reviewer's high level point is to make sure that we present results in a way that prioritize scenarios where our methods can show significant performance gains, hence a better way to convince readers of our methods' superiority. We agree that this would be a potentially good strategy for presentation in some sense.
> >
> > In our presentation, we did not optimize for the ordering in which the scenarios (unverifiable vs. verifiable) are presented, mainly because we feel it is a more fair and forthright approach to showcasing the results. We would like to be transparent about settings in which JEPO falls behind slightly (i.e., the verifiable case where RLVR is supposed to dominate), or even to emphasize the key trade-offs here, so that practitioners might find such results more genuine and practically relevant.
> >
> > > Your efforts are probably better spent focusing on demonstrating where your method is advantaged (and justifying why those situations where your method is advantageous are of practical relevance/importance).
> >
> > Indeed, we have put much effort on the performance gains in the semi-verifiable and unverifiable case would convince readers of the practical trade-offs involved in the application of JEPO vs. RLVR vs. SFT. Since RL on verifiable domain has sparked much research in the field recently, and understanding better how to apply RL to the unverifiable domain is on the horizon, we hope our results offer insights mainly along this line.

---

> > > ### Comment · Reviewer_kgcS · 2025-08-03
> > >
> > > Thanks for your response!
> > >
> > > This will likely be my last response.
> > >
> > > > In our presentation, we did not optimize for the ordering in which the scenarios (unverifiable vs. verifiable) are presented, mainly because we feel it is a more fair and forthright approach to showcasing the results. We would like to be transparent about settings in which JEPO falls behind slightly (i.e., the verifiable case where RLVR is supposed to dominate), or even to emphasize the key trade-offs here, so that practitioners might find such results more genuine and practically relevant.
> > >
> > > I do appreciate honesty and transparency, as I think this is lacking in many papers (including published and even some fairly high profile papers). There are too many papers out there that claim improvements which end up not really being true, due to hacks, poor baselines, poor evaluations, cherry picking, bugs, etc. Still, I think the presentation could be improved.
> > >
> > > Overall, I think if all of the promised changes above were implemented (and no key results were challenged as a result of this; e.g., additional seeds with confidence intervals were reasonable and did not materially affect conclusions), I would be happy to give a "borderline accept". As it stands, while the authors' responses have slightly improved my evaluation of the paper, I'm still on the fence and am not sure I would increase my evaluation score at present.

---

> > > > ### Author Response · Authors · 2025-08-05
> > > > **Thank you for your reply!**
> > > >
> > > > Thank you for the continuous engagement with the discussion.
> > > >
> > > > > I do appreciate honesty and transparency, as I think this is lacking in many papers (including published and even some fairly high profile papers). There are too many papers out there that claim improvements which end up not really being true, due to hacks, poor baselines, poor evaluations, cherry picking, bugs, etc. Still, I think the presentation could be improved.
> > > >
> > > > Thank you for the recognition on this point. Indeed, we agree that there is room for improvement on the presentation quality, and we indeed believe that honesty and transparency better serves the community in the long run.
> > > >
> > > > > Overall, I think if all of the promised changes above were implemented (and no key results were challenged as a result of this; e.g., additional seeds with confidence intervals were reasonable and did not materially affect conclusions), I would be happy to give a "borderline accept". As it stands, while the authors' responses have slightly improved my evaluation of the paper, I'm still on the fence and am not sure I would increase my evaluation score at present.
> > > >
> > > > Thank you for all your suggestions above. We will implement the changes as you suggested in the manuscript of the paper (presentation, typos, additional discussions), as part of the camera ready version if we could come to that stage. Indeed, as the reviewer noted this will not change the results of this work in a material way, and we certainly hope that our response has helped convince you more of the significance of this work.
> > > >
> > > > Please let us know if you have any other specific questions.

---

> > > > > ### Comment · Reviewer_kgcS · 2025-08-06
> > > > >
> > > > > Thanks for your continuous engagement as well!
> > > > >
> > > > > One final note:
> > > > >
> > > > > >  Indeed, as the reviewer noted this will not change the results of this work in a material way
> > > > >
> > > > > That's not what I meant. I meant if the changes above were implemented and did not affect the conclusions, I would be happy. I did not say that the changes would not affect the results. That remains to be seen (as there are a few empirical items).

---

> > > > > > ### Author Response · Authors · 2025-08-06
> > > > > > **Thank you for the final note!**
> > > > > >
> > > > > > Thank you very much for the clarification and your continuous engagement.
> > > > > >
> > > > > > > That's not what I meant. I meant if the changes above were implemented and did not affect the conclusions, I would be happy. I did not say that the changes would not affect the results. That remains to be seen (as there are a few empirical items).
> > > > > >
> > > > > > We sincerely apologize for misinterpreting the previous message. We meant to say that we speculate that the additional empirical items hopefully would not alter the main conclusion of this work. For example, though admittedly we do not have multiple seed runs as in a typical deep RL paper, if we examine results across the board, the trends of the gains are reasonably consistent across different experiments (see, e.g., Fig 2-4).
> > > > > >
> > > > > > Though multiple seeds will further solidify the conclusion, there is a practical trade-off to be made here since LLM RL runs are considerably more costly (and most existing papers present single runs as a result of this). That said, we understand the general points made by the reviewer here.
> > > > > >
> > > > > > Many thanks please let us know if you have any other questions or thoughts.

---

### Official Review · Reviewer_SjMC · 2025-07-05

**Clarity:** 4
**Significance:** 4
**Originality:** 4
**Rating:** 5
**Confidence:** 5

**Summary:**

The paper introduces a new rl scheme for training on datasets where prompts and ground-truth answers are available, but the correctness of generated solutions is difficult to verify. The method aims to maximize the likelihood of the ground truth given the prompt, treating the cot reasoning as a latent variable. A Jensen lower bound is used as a tractable surrogate for the likelihood. Several techniques and theoretical analyses are proposed to tighten the bound and reduce variance. Comprehensive experiments are conducted to validate the approach.

**Questions:**

- **Entropy dynamics**: In JEPO, how does the entropy behave during training? In standard verifiable RL training, entropy typically decreases and then stabilizes. Does JEPO follow a similar pattern, or is the behavior different? Would adding an explicit entropy loss to JEPO be beneficial?

- **Answer granularity**: In the verifiable reward setting, is the correct answer $a^*$ just the final boxed answer (e.g., an integral value in AIME problems), or does it include the full solution reasoning block? How do you extract $c_i$ from the generation? Do you simply parse the content inside `<think>` tokens?

- **Robustness to incorrect labels**: How robust is JEPO to incorrect ground-truth labels $a^*$? For example, in [1], it was shown that RLVR is relatively robust to such label noise. Have you observed similar robustness in JEPO?

[1] Shao, Rulin, et al. *Spurious rewards: Rethinking training signals in RLVR*. arXiv preprint arXiv:2506.10947 (2025).

**Ethical Concerns:**

["NO or VERY MINOR ethics concerns only"]

**Final Justification:**

The method is interesting and statistically sound. I like the paper and would be glad to see it accepted.

**Limitations:**

No obvious limitations are observed.

**Quality:**

4

**Strengths And Weaknesses:**

**Strengths:**

- The problem of extending RL from verifiable to unverifiable settings is important and timely.
- The proposed method is theoretically sound. The techniques introduced to tighten the lower bound and control variance are solid.
- The experimental evaluation is thorough, covering verifiable, semi-verifiable, and unverifiable settings, and scaling across models from 7B to 70B parameters. The method demonstrates strong generalization and scalability.
- The ablations on sample count $n$ and the sft coefficient are insightful.

**Weaknesses:**

- I do not observe any obvious weaknesses in the paper.

---

> ### Author Rebuttal · Authors · 2025-07-29
>
> Thank you very much for your time and efforts in reviewing our manuscript. We will make sure to incorporate your feedback into our later revisions.
>
> Thank you very much for your constructive feedback and very positive view towards our paper!
>
> > Entropy dynamics: In JEPO, how does the entropy behave during training? In standard verifiable RL training, entropy typically decreases and then stabilizes. Does JEPO follow a similar pattern, or is the behavior different? Would adding an explicit entropy loss to JEPO be beneficial?
>
> In JEPO, entropy dynamics behaves rather similarly as regular RL training. This might be because JEPO ultimately looks to find CoTs which maximize the marginal likelihood of the ground truth answer, in a similar spirit as regular RL that finds CoTs that make the ground truth answer generation more likely.
>
> We speculate that adding entropy to JEPO would potentially have a similar effect as adding entropy to regular RL. It might help avoid premature convergence in some cases but tuning the entropy coefficient itself involves some hyper-parameter tuning to make things stable.
>
> > Answer granularity: In the verifiable reward setting, is the correct answer  just the final boxed answer (e.g., an integral value in AIME problems), or does it include the full solution reasoning block? How do you extract  from the generation? Do you simply parse the content inside <think> tokens?
>
> Thank you for raising this point! We have actually discussed the specifics of this important technical detail in Appendix B. We will make it more clear in the revision.
>
> During SFT we have taught the model to carry out reasoning, followed by the final answer is, followed by the ground truth answer (e.g., an integer value). In other words, the overall string output from the model should be
>
> Reasoning + `the final answer is` + answer
>
> We extract the string before the marker `the final answer is` as the reasoning CoT $c_i$, and the string after as the final answer $a$. If the model does not comply with this formatting, we penalize the model output with a constant penalty. We find that with this, the model can quickly learn to format the full response correctly.
>
> Another idea is indeed as the reviewer suggests, we can use a special thinking token as a delimiter, rather than a marker string. This will potentially make the formatting more stable and the parsing more convenient.
>
> > Robustness to incorrect labels: How robust is JEPO to incorrect ground-truth labels ? For example, in [1], it was shown that RLVR is relatively robust to such label noise. Have you observed similar robustness in JEPO?
>
> We have not thoroughly investigated the robustness of JEPO performance to ground truth labels. Though based on the conceptual connection between JEPO and regular RLVR, we would expect some robustness to transfer given the evidence in [1]. It would be a valuable follow up ablation to carry out, to better understand whether JEPO scales better in the presence of data noise.

---

> > ### Comment · Reviewer_SjMC · 2025-08-05
> >
> > Thank you for the authors’ response. I have a few minor questions:
> >
> > 1.As noted in the appendix, regarding the loss in line 615: should the gradient term in the first component be $\nabla_\theta \log \pi_\theta(c_i \mid x)$ instead of $\nabla_\theta \log \pi_\theta(a_i \mid x, c_i)$? Otherwise, since $a_i$ is the generated answer, it could potentially be incorrect and introduce noise.
> >
> > 2.Does the length or format of $a^\*$ affect training efficiency or final performance? For example, in AIME-style datasets, should $a^\*$ include the full block of reasoning steps along with the final answer, or just the final answer (e.g., an integer)? Intuitively, one might expect the former to provide more informative supervision—should it lead to better performance than the latter?
> >
> > 3.Will the code be made publicly available?

---

> > > ### Author Response · Authors · 2025-08-05
> > > **Thank you for your reply!**
> > >
> > > Thank you very much for your time in replying to our rebuttals, we appreciate your time very much.
> > >
> > > > 1.As noted in the appendix, regarding the loss in line 615: should the gradient term in the first component be $\nabla_\theta \log \pi_\theta(c_i \mid x)$ instead of $\nabla_\theta \log \pi_\theta(a_i \mid x, c_i)$? Otherwise, since $a_i$ is the generated answer, it could potentially be incorrect and introduce noise.
> > >
> > > Your comment is correct and we apologize for the typo! Indeed, it should have been $\log \pi(c|x)$ rather than the action, since this part of the loss is meant to optimize the CoT. Also indeed as the reviewer noted, the second part of the loss is also always with respect to the ground truth answer $a^\ast$.
> > >
> > > > 2.Does the length or format of
> > >  affect training efficiency or final performance? For example, in AIME-style datasets, should
> > >  include the full block of reasoning steps along with the final answer, or just the final answer (e.g., an integer)? Intuitively, one might expect the former to provide more informative supervision—should it lead to better performance than the latter?
> > >
> > > This is definitely a meaningful practical concern. For AIME-style problems, where the *canonical* ground truth answer is short-form (just a short string representing the integer valued answer), we take $a^\ast$ as the short answer. We understand that dataset such as the AIME training set comes with golden step-by-step solution too, which we repurpose during SFT to warm-start RL/JEPO, rather than during RL/JEPO.
> > >
> > > For the unverifiable dataset such as Numina-proof, we take $a^\ast$ as the full step-by-step reasoning of proof. In this case, the step-by-step proof is the answer itself, as there is not a *single* answer at the end of the response. Here, the chain-of-thought $c$ the model outputs, tends to be a high-level strategy that outlines the proof, which makes the final step-by-step proof more likely.
> > >
> > > Finally, regarding your primary comment: the length of the answer $a^\ast$ definitely impacts the learning dynamics significantly. When $a^\ast$ is short-form, intuitively, predicting it from a long CoT $c$ is more straightforward than when $a^\ast$ is long-form. As a result, the supervised learning loss component $\log \pi(a^\ast|c)$ plays a bigger role when $a^\ast$ is long-form.
> > >
> > > > 3.Will the code be made publicly available?
> > >
> > > Unfortunately we potentially won't make the code public due to a few constraints, we apologize for the inconvenience from this! That said, we have tried to be as concrete as possible with respect to key implementation designs (Appendix B.2) and believe that it is relatively straightforward to incorporate the changes into existing RL/RLVR pipelines.
> > >
> > > Please let us know if you have any other specific questions, thanks!

---

> > > > ### Comment · Reviewer_SjMC · 2025-08-08
> > > >
> > > > Thank you for the authors' response—all of my concerns have been addressed. I'm glad to see the paper accepted. Since my rating is already high, I will keep it as is.

---

### Note · Authors · 2025-08-14

We are encouraged by the initial positive reviews from most of the reviewers. We also benefit from the follow-up discussion with reviewer SjMC, kgcS and roE2 which will help improve our final manuscript. Our discussion with reviewer kgcS has been the most extended, and while we understand that kgcS has doubts with our work, we thank the reviewer for the continuous engagement and suggestions raised. Here, we would like to to close with a few high-level points in the discussion regarding main contentions from the reviewer.

- Reviewer kgcS raised many points on the improvement of presentation. While we acknowledge the room for improvement, which we will act upon in the revision when the space limit is slightly alleviated, we want to highlight our intent of clarity and transparency. Crucially, we do not frame JEPO as a "new algorithm that beats all SOTA", instead, it is a training paradigm with pros and cons. Indeed, we have shown that JEPO, while still competitive, might not take the full advantage of a good verifier in the verifiable domain compared to RLVR baseline. Yet very important, in the unverifiable domains, which is the research frontier, we see strong improvements. In practice, algorithmic choices must be made depending on applications and data, and we believe practitioners will generally appreciate such a presentation of results.

- Reviewer kgcS raised the point on confidence interval. While we acknowledge the importance of empirical results with uncertainty calibration, we also want to point out that due to the expensive nature of LLM experiments, most papers in the space do not carry out multi-seed study, quite unlike the deep RL "traditions". Instead, compute is spent on various ablations and different data setting. We believe confident conclusions can still be drawn in view of the whole experimental design.

- A few reviewers raised the need for a calibrated judge score. We agree this is a good point to bring more clarity over in the final manuscript: our preliminary validation of judge + Sympy scorer, shows that the combined score yields more accurate evaluation assessment than the Sympy scorer alone (due to improve false negative rates), bringing more confidence into our automatic evaluation process. We will provide more details in the revision.

Finally, we thank the reviewers and AC again for their time and valuable suggestions.

---

### Decision · Program_Chairs · 2025-09-17

**Decision:**

Accept (poster)

**Comment:**

This paper addresses the problem of scaling reinforcement learning for LLMs beyond verifiable data, where correctness cannot be automatically checked (e.g., mathematical proofs). The authors propose JEPO (Jensen’s Evidence lower bound for Policy Optimization), which treats chain-of-thought as a latent variable and maximizes a Jensen-tightened lower bound on the log-likelihood of ground-truth answers. This enables RL training without explicit reward signals, covering verifiable, semi-verifiable, and unverifiable domains. Experimental results are also reported.

Reviewers agreed that the problem is timely and found the method theoretically sound and practically relevant. Strengths highlighted include the conceptual novelty of reframing RL as latent-variable modeling, clear applicability to unverifiable reasoning tasks, and comprehensive ablations across different datasets and model sizes. The paper suffers from some presentation issues (typos, figure placement, lack of clarity in some derivations), limited evaluation breadth (primarily math-based tasks, broader OOD tests or stronger verification baselines), and reliance on proxy metrics like LLM-as-a-judge for unverifiable data.

Overall, I recommend acceptance. The contribution is technically solid and addresses an important gap in reinforcement learning for language models, especially in unverifiable domains where existing RL methods fall short. While presentation and empirical breadth can be improved, the rebuttal clarified key concerns and the main results remain sound. Please incorporate the final suggestions of the reviewers in the final version as promised.